# All-metallic magnetic Purcell enhancement in a thermally stable room-temperature maser

Rongrong Xiang ⬡ , Philippe Bugnon, Maliheh Khatibi Moghaddam & Romain Fleury ⬡ ✉

Enhancing the spontaneous and stimulated emission rates of magnetic quantum emitters through the Purcell effect is essential for designing high-performance quantum devices based on high quality factor and small mode volume. At room temperature, structures constructed using high-index dielectric materials have been favored due to their ability to effectively confine electromagnetic fields. However, these dielectric resonators are plagued by an inherent sensitivity to thermal variations, which are unavoidable during the optical excitation or readout of the quantum states. Here, we propose to solve this issue with a dielectric-free, all-metallic toroidal split-ring resonator cluster of subwavelength size ($\leq \lambda_m/17$), which exhibits a fundamental magnetic mode at approximately 1.45 GHz ($\lambda_m \sim 207$ mm) and a remarkably low mode volume ($8.1 \times 10^{-6} \lambda_m^3$). Through experimental investigations, we demonstrate that the proposed resonator exhibits a high Purcell factor ($5 \times 10^6$), and observe maser action when paired with a pentacene-based gain medium. We evidence the remarkable stability of the output pulse against thermal variations caused by thousands of consecutive optical excitations, by far surpassing that of masers based on dielectric resonators.

Spin-based magnetic transitions have recently attracted a significant amount of attention in the context of quantum information and sensing technologies[1–3], due to their potentially long coherence lifetimes, their ability to be tuned using external magnetic fields, and the possibility to optically control level population using spin-selective transitions, even at room temperature. Despite their intrinsic weakness in natural materials[4], magnetic transitions can be obtained in particular systems, including rare-earth ions[5], quantum dots[6], nitrogen-vacancy (NV) centers in diamond[7] and organic molecules like pentacene[8]. Controlling the emission rate of such spin-based magnetic transitions is thus pivotal for a wide range of applications, including quantum computing[9,10], quantum sensing[11,12], electron-spin resonance spectroscopy[13], and active emission devices[14].

The Purcell effect, which describes the fundamental back-action of the environment on the dynamics of quantum emitters, is a key phenomenon to control emission rates in quantum electrodynamics[15].

As shown by Purcell in his seminal paper[16], the spontaneous emission rate of quantum emitters can be enhanced when placed in a cavity with a high quality factor ($Q$ factor) and a small mode volume ($V_m$). This enhancement, characterized by the Purcell factor ($F_p \propto Q\lambda_m^3/V_m$), equivalently contributes to the stimulated emission process[17]. However, as spin transition frequencies are commonly in the gigahertz range, i.e., with emission wavelengths ($\lambda_m$) on the order of tens of centimeters, leveraging the magnetic Purcell effect comes with the inherent challenge of dealing with the large ratio between $\lambda_m^3$ and achievable values of the mode volume $V_m$. In practice, the physical dimensions of gain media that can be grown experimentally, such as organic crystals or diamond-based samples, occupy physical volumes that are roughly three orders of magnitude smaller than $\lambda_m^3$ (i.e., $\propto$ cm³). To address this mismatch, the resonator geometry must be engineered such that the region of strong magnetic field is confined to a volume only slightly larger than that of the gain medium, with its

Laboratory of Wave Engineering, École Polytechnique Fédérale de Lausanne, Lausanne, Switzerland. ✉e-mail: romain.fleury@epfl.ch

orientation optimized for alignment with the emitters' magnetic dipole moments. This squeezing of the magnetic mode maximizes spatial overlap with the active medium, thereby enhancing the efficiency of energy exchange.

To circumvent this issue, many studies have instead focused on increasing the quality factor $Q$, for example, by using superconducting materials[13,18], at the cost of operating under cryogenic conditions. At room temperature, small mode volumes and high $Q$ have been achieved by employing dielectrics with very high permittivities and/or low dielectric losses, such as strontium titanate ($SrTiO_3$)[19] and sapphire[20,21]. These approaches have enabled a substantial enhancement of magnetic stimulated emission, sufficient to achieve maser action. However, dielectric resonators with extremely high permittivities, such as $SrTiO_3$, are inherently sensitive to temperature variations due to their large temperature coefficient of permittivity. Even minor room-temperature variations, or the unavoidable thermal heating coming from optical excitation or readout[22], would inevitably shift the mode frequency away from the quantum emitter linewidth. As a result, the system must be continuously retuned, hindering stable operation and limiting the practicality of such dielectric resonators.

In this study, we fix this vexing issue and demonstrate that room-temperature magnetic quantum emission enhancement can be stable over hours in a metamaterial-inspired, all-metallic magnetic resonator with a remarkably small mode volume, five orders of magnitude smaller than $\lambda_m^3$. This is obtained with a specially designed multiple three-dimensional split-ring resonator (SRR) cluster, monolithically manufactured by lost-wax metal casting, which is an indirect additive manufacturing technique. We provide direct evidence that this resonator significantly boosts the stimulated emission rate of magnetic quantum emitters, by constructing a room-temperature pentacene maser that is completely free of high-index dielectric materials. We demonstrate maser operation stable in frequency and in output pulse shape over thousands of seconds, which is not achievable in prior dielectric-based designs.

## Results

### Ultra-strong magnetic Purcell-enhanced stimulated emission

Planar split-ring resonators have been widely used as metal-based resonant magnetic scatterers in metamaterials and metasurfaces[23],

where they provide a compact way for strong subwavelength magnetic interactions and dipolar emission control[24]. We have devised here a complex three-dimensional cluster of such resonators comprising four concentric split rings (see the inset in Fig. 1a), whose geometry has been optimized through full-wave finite-element simulations to yield a strong spatial confinement of magnetic energy in its middle region. The SRR cluster is shaped as a torus with an inner radius of 1.52 mm, outer radius of 6.02 mm and a height of 4.5 mm (other geometrical details are shown in Supplementary Fig. S1). To facilitate monolithic fabrication and avoid post-fabrication alignment steps, small metallic connections were added to link adjacent split rings within the cluster, with diameters of 1 mm. These interconnections have been located at adequate positions, so that they do not significantly influence the magnetic resonance. Additionally, the outermost ring is securely fixed to a circular stand via two thin posts having 1-mm diameter. This all-metal monolithic SRR cluster has a fundamental magnetic resonant mode at approximately 1.45 GHz, with a high concentration of magnetic energy at its core (1.5 mm radius × 4.5 mm height), and a magnetic field mostly oriented along the $z$-axis, as shown in Fig. 1a (the electric field distribution is shown in Supplementary Fig. S2). Assuming that within this region, we align uniformly an ensemble of quantum magnetic emitters along the $z$-axis, we would get an effective magnetic mode volume $V_m = \int \frac{|\mathbf{H(r)}|^2}{|\mathbf{H_{max}(r)}|^2} dV$ of only 0.025 cm³. Remarkably, this estimation is almost 350 thousand times smaller than $\lambda_m^3$. Such a resonator has an unloaded $Q$ factor ($Q_{unload}$) around 700, assuming that it is fabricated in silver and enclosed within an copper cavity to reduce the radiation losses. The corresponding single-mode Purcell factor $F_p = \frac{3\lambda_m^3}{4\pi^2} \cdot \frac{Q}{V_m}$ [16] is estimated to be around $1.89 \times 10^7$, which is of the same order of magnitude as the one reported using strontium titanate, a record-high dielectric with $\epsilon_r \sim 320$[25]. Our all-metallic design is also significantly different from metallic structures discussed in other works[26–29]. In the following, we perform a direct magnetic quantum emission experiment to establish the relevance of our design.

In light of these promising simulation results, we have selected the pentacene-based pulsed maser[20] as an ideal test platform, for which we can directly compare the performance obtained with our SRR cluster and with a high-index dielectric resonator. This particular maser is

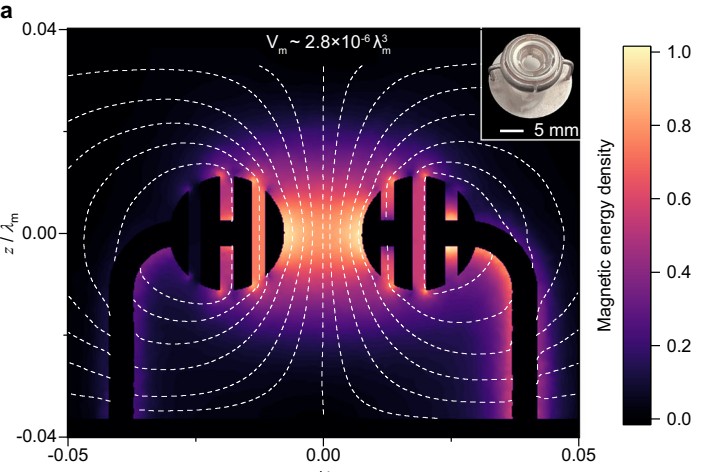

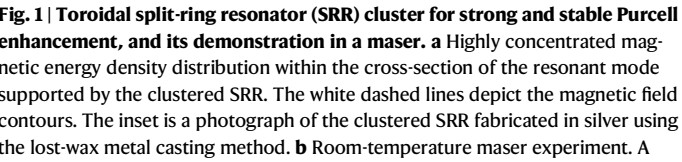

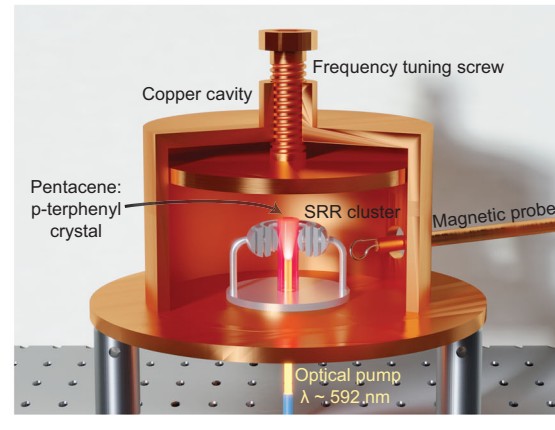

**Fig. 1 | Toroidal split-ring resonator (SRR) cluster for strong and stable Purcell enhancement, and its demonstration in a maser. a** Highly concentrated magnetic energy density distribution within the cross-section of the resonant mode supported by the clustered SRR. The white dashed lines depict the magnetic field contours. The inset is a photograph of the clustered SRR fabricated in silver using the lost-wax metal casting method. **b** Room-temperature maser experiment. A

pentacene:p-terphenyl crystal is loaded into the clustered SRR. Both are housed inside a cylindrical copper cavity featuring an adjustable copper top, which enables adjusting the resonance frequency. The pentacene molecules are excited by a yellow pulsed laser via an optical fiber inserted into the crystal through the cavity bottom. A loop probe is utilized to extract the maser signals.

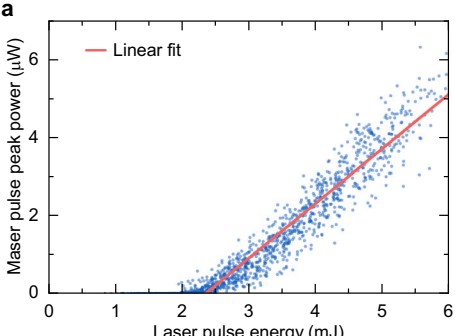
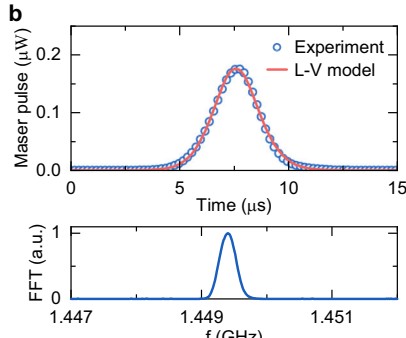
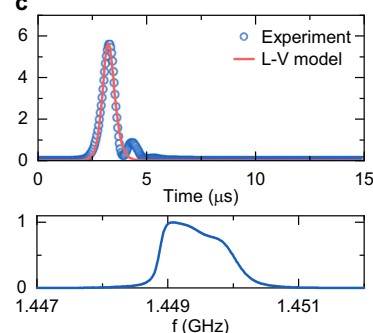

**Fig. 2 | Experimentally measured output pulses from the SRR-cluster maser.**
**a** Peak power of the maser pulses as a function of the pump laser energy. The linear fit (pink line) of the measured data (blue dots) suggests that the pump threshold of the maser is around 2.3 mJ. **b**, **c** Maser pulses in time domain (top) and corresponding fast Fourier transformed (FFT) spectra (bottom) under laser excitation of 2.5 mJ and 5.9 mJ, respectively. The measured maser pulses (blue circles) are fitted using the L-V model (pink line). With higher pump energy, the maser pulse exhibits a weak oscillatory behavior and corresponding spectral broadening.

distinguished by its ability to achieve room-temperature masing without requiring an external magnetic field. The maser action predominantly results from the population inversion (N) established between the X and Z sublevels within the first triplet state of pentacene molecules under optical excitation (see Supplementary Fig. S3a)[30,31]. As depicted in Fig. 1b, the pentacene:p-terphenyl (pc:pt) crystal (1.5 mm radius × 10 mm height) is positioned within the bore of the clustered SRR, which is enclosed inside a copper cavity. The bottom 5 mm of the pc:pt crystal includes a central axial hole of 0.5 mm radius, whereas the top 5 mm remains solid. The resonant frequency of the cavity can be manually fine-tuned to approximately 1.4493 GHz using the top tuning plate. To efficiently pump the crystal, the laser pulses (wavelength at 592 nm, pulse duration ~ 5 ns, repetition rate 1 Hz) from an optical parametric oscillator (OPO, SL I-20 Surelite) are coupled into an optical fiber (core diameter 1 mm) with a coupling efficiency ~ 60%. The other end of the optical fiber protrudes through the cavity bottom and is inserted into the central hole of the crystal. The maser burst following each laser pulse is then captured using a weakly coupled magnetic loop probe and recorded by an oscilloscope (Tektronix MSO64-2.5G). A schematic of the complete experimental setup can be found in Supplementary Fig. S4.

By gradually increasing the pulse energy of the pump laser from 0.8 mJ to 6 mJ, we recorded a total of 1300 maser pulses and analyzed them using a Lotka–Volterra model (L-V model, see "Methods")[32,33]. A collection of the peak power in all maser pulses as a function of the laser pulse energy indicates that there is an optical pumping threshold around 2.3 mJ, below which no maser emission can be detected (Fig. 2a). The slight fluctuations in the maser output are a consequence of fluctuations in OPO output beams (see Supplementary Fig. S5). Two maser pulses under excitation energy of 2.5 mJ and 5.9 mJ are given as instances in Fig. 2b and c, respectively. When the excitation energy slightly exceeds the threshold, the maser pulse possesses a single peak in time domain and can be accurately fitted by the L-V model, yielding an initial population inversion $N \sim 5.2 \times 10^{13}$. With pumping energy well above the threshold ($N \sim 6.8 \times 10^{13}$), we observe a weak oscillatory behavior in the maser pulse. This behavior could potentially arise from coherent energy exchange between the spin ensemble and the cavity mode[34], or from collective emission phenomena such as superradiance[35,36]. Further experimental and theoretical investigations will be necessary to determine the precise underlying mechanism of these oscillations.

### Excellent spectral robustness against thermal variations

To demonstrate the thermal stability of the proposed clustered SRR, we consecutively recorded the temporal responses of the maser to laser excitation over a period of five hours. The successive laser excitation, operating at a 1-Hz repetition rate with pulse energy around 3 mJ, induced thermal heating in the resonator and the pc:pt crystal, since a given amount of the optical energy is dissipated without contributing to the masing process. As a reference, we built a second maser using a standard hollow cylindrical strontium titanate (SrTiO$_3$) resonator, which is similar in size to our clustered SRR and exhibits a comparable Purcell factor. The SrTiO$_3$ resonator possesses a high Purcell factor ($F_p \sim 3.6 \times 10^7$), thanks to its high relative permittivity and low loss tangent ($\epsilon_r \sim 320$, $\tan(\delta) \sim 9 \times 10^{-5}$)[19]. The emission spectra of all the pulses, measured under the same conditions from the clustered-SRR- and SrTiO$_3$-based masers, are plotted in Fig. 3a, b, respectively. It is evident that the spectra of the clustered-SRR-based masers exhibit a superior stability in both signal strength and resonant frequency over at least 18,000 shots. This excellent thermal stability is attributed to silver's high thermal conductivity and moderate thermal expansion coefficient, resulting in a temperature coefficient of resonant frequency of approximately 130 ppm/K. In contrast, the central frequency of the SrTiO$_3$-based maser shifts from 1.449 GHz to 1.453 GHz, resulting from SrTiO$_3$'s large temperature coefficient of resonant frequency (approximately 1700 ppm/K)[25,37]. We simulated the diffusion of the deposited optical pump energy and the resulting temperature variations in the SrTiO$_3$ resonator using COMSOL. The simulation results are presented in the Supplementary Movie 1 and Fig. S6. More importantly, this frequency shift is accompanied by a variation in the signal strength, which eventually vanishes because of the limited gain linewidth of pentacene molecules (see Supplementary Fig. S3b). We terminated the measurement in the SrTiO$_3$-based maser after 12,000 shots as no maser signal was detected during the final 200 s. After switching off the pump laser, the resonant frequency of the SrTiO$_3$ resonator took approximately two hours to recover back to its initial value (see Supplementary Fig. S7). Apart from the self-termination, another side effect caused by the frequency shift is the transition in the maser operation mechanism: it enters the strong-coupling regime featuring a oscillatory response, and later returns to the Purcell-enhanced regime (see Fig. 3d).

## Discussion

The L-V model yields an Einstein coefficient of stimulated emission, $B \approx 38.7 \times 10^{-8}\,\text{s}^{-1}$, which is about 3.5 times higher than the value reported for the SrTiO$_3$-based maser[19,32]. From this value, the experimentally inferred mode volume of the clustered SRR is approximately $V_m \approx 0.072\,\text{cm}^3$, which is nearly three times larger than that obtained from the numerical simulation. Since $V_m \propto \frac{1}{|\mathbf{H} \cdot \mathbf{n}|^2}$[38], where **H** denotes the magnetic field of the cavity mode and **n** is the unit vector pointing in the direction of the magnetic dipole moment, this discrepancy can

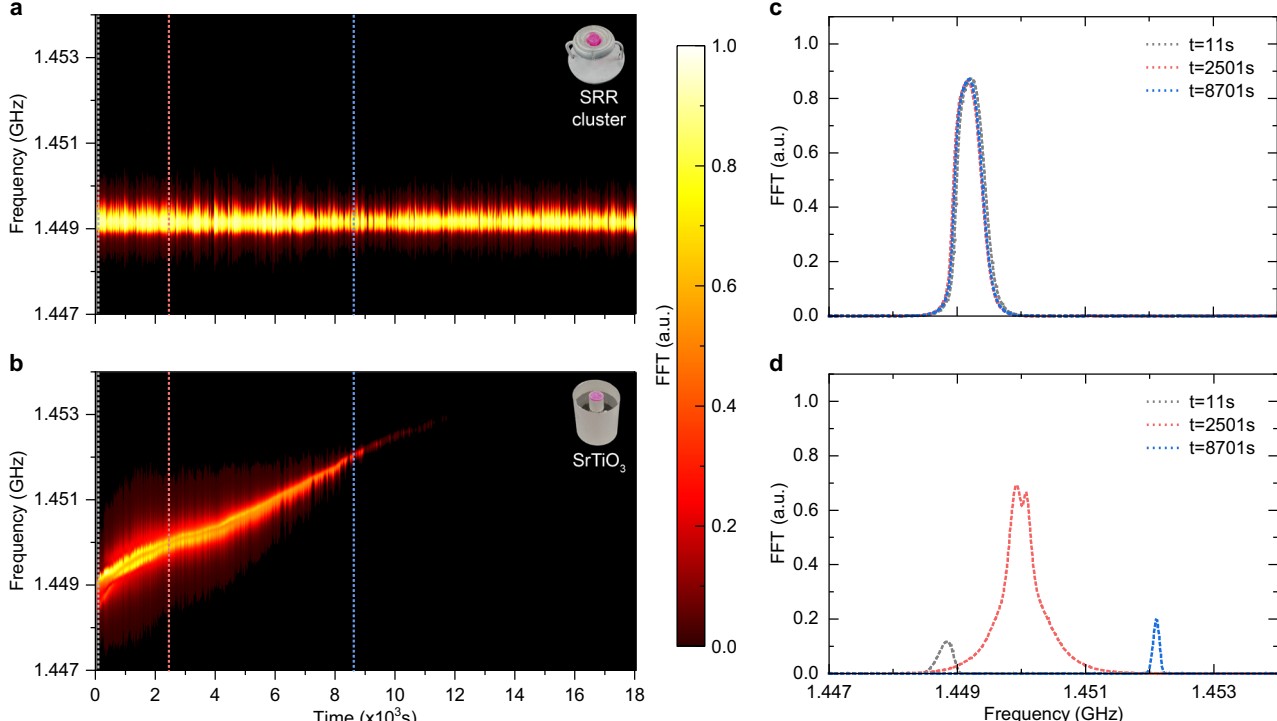

**Fig. 3 | Stability of operation of the clustered-SRR-based maser over a duration of 18,000 s. a, b** Emission spectra of masers constructed with the SRR cluster and SrTiO₃ resonator, respectively. Both masers were experimented under the same conditions and pumped by the OPO laser at a 1-Hz repetition rate (pulse energy around 3 mJ). Spectra of the clustered-SRR-based maser remained consistently stable throughout the entire experiment. In contrast, spectra of the SrTiO₃-based maser gradually diminished with a blueshift in its central frequency. **c, d** Selected emission spectra from the clustered-SRR- and SrTiO₃-based masers, respectively, at time 11, 2501, and 8701 s. The clustered-SRR-based maser operates within the same physical regime at all times, whereas the SrTiO₃-based maser underwent a dynamic transition into the oscillatory regime and subsequently reverted to the Purcell-enhanced regime due to its shifted resonant frequency.

be attributed to the misalignment between the magnetic dipole emitters and the magnetic field of the cavity mode. Potential sources for this misalignment include: (i) geometric deformation of the clustered SRR caused by the metal casting technique; and (ii), structural defects in the pentacene:p-terphenyl crystal that introduce variations in dipole orientation. To confirm that the simulated mode volume is not artificially reduced by local field maxima, we quantitatively analyzed the magnetic energy distribution in the resonator bore region. The field was found to be broadly uniform without strong localization, as detailed in Supplementary Discussion SII A. With $V_m = 0.072$ cm³ and $Q_{load} = 530$, the experimental Purcell factor of the fabricated SRR cluster is calculated to be $5 \times 10^6$. Furthermore, to assess the interaction strength between the emitters and the confined EM mode in the resonator, the cooperativity of the maser, constructed with the SRR cluster, can be calculated using $C = 4g^2 N/k_c k_s$, where $k_s = 2/T_2^*$ is the spin dephasing rate with the spin dephasing time $T_2^* \approx 2.9\,\mu s$ and $g = \gamma_e \sqrt{\mu_0 \hbar \omega / 2V_m}$ is the single spin-photon coupling strength. With $N = max(N_0) = 7.3 \times 10^{13}$, the cooperativity is estimated to be $C \approx 6.4$, suggesting a strong coupling between the emitters and the SRR cluster.

In many cases, optimization of the Purcell factor involves making compromises between the $Q$ factor and the mode volume. Cavities with a high $Q$ would be beneficial for applications such as reducing the emissive threshold of active devices (e.g., lasers and masers, etc.). However, when the emitter's transition linewidth is much larger than the linewidth of the cavity mode ($\Delta\omega_e \gg \Delta\omega_c$), the emission rate is no longer mainly modified by $\frac{\omega_c}{\Delta\omega_c} \cdot \frac{1}{V_m}$. Instead, it becomes proportional to $\frac{\omega_c}{\Delta\omega_e} \cdot \frac{1}{V_m}$, thus limiting the Purcell enhancement introduced by the high $Q$ factor[17,39]. On the other hand, $V_m$ still contributes to the Purcell effect

without being hindered by the emitter/cavity relative linewidth. Despite having a modest $Q$ factor, our proposed SRR cluster provides a strong Purcell enhancement thanks to its ultra-small $V_m$. Consequently, it imposes fewer restrictions on the spectral linewidth of emitters. Furthermore, as $Q$ factors are inherently limited by material properties, engineering the effective mode volume of the environment is particularly advantageous in numerous applications.

In conclusion, we have experimentally demonstrated exceptionally stable maser operation using a specially designed all-metallic SRR cluster, reaching a high Purcell factor of approximately $5 \times 10^6$ at room temperature. It possesses a deep-subwavelength magnetic mode volume ($V_m = 0.072$ cm³ $\sim 8.1 \times 10^{-6} \lambda_m^3$), meanwhile provides a considerable space for positioning and aligning the dipole emitters in the region where the magnetic energy is most concentrated. Remarkably, it is resilient to the thermal variations occurring under prolonged laser excitation. For future work, we surmise that the Purcell factor of this resonator could be further improved, especially through the use of more precise fabrication techniques and by improving our control over magnetic dipole orientations and distribution in the gain medium. In addition, the geometry of the clustered SRR can be tailored to achieve operation at different frequencies, accommodating other magnetic quantum emitters such as NV defect centers in diamond, which are used in continuous-wave (CW) masers[21]. A detailed analysis of the thermal stability of CW masers incorporating the SRR cluster is provided in Supplementary Discussion SII B. Moreover, the utilization of quantum emitters in high concentration could promote light-matter interaction into the strong-coupling regime. Our study demonstrates the possibility to avoid the drawbacks associated with the use of high-dielectrics in room-temperature magnetic quantum technologies.

## Methods

### Design and fabrication of the clustered SRR

The resonator was designed using eigenmode and full-wave frequency-domain solvers in CST Studio Suite for obtaining the optimal magnetic Purcell factor. This design trajectory is concisely summarized in Supplementary Fig. S8. The optimized toroidal SRR cluster was monolithically printed in sterling silver using wax 3D printing and lost-wax metal casting (3D Print India). The 3D model of the clustered SRR is available on the repository Zenodo (see "Data availability") for precise reproduction of the structure. No post-fabrication alignment was needed thanks to its monolithic structure. The printed SRR cluster was electroplated with 15–20 μm thick silver (Steiger galvanotechnique SA) to reduce the ohmic losses.

### Pentacene:p-terphenyl crystal growth

The pentacene:p-terphenyl crystal (concentration around 0.1% mol/mol) is grown using the vertical Bridgman method. Commercially purchased p-terphenyl (A14833, Alfa Aesar) is purified by sublimation under high vacuum ($\leq 10^{-6}$ mbar), and then mixed with the pentacene (P2524, TCI Europe NV) powder as purchased. For the purpose of creating a central hole through the crystal, a pyrex rod (diameter 1 mm) is inserted after loading the mixed powders into a doubled-layer Pyrex ampoule (inner diameter 3 mm, Tecglas), which is then sealed under vacuum ($\leq 10^{-6}$ mbar). The ampoule has a tapered capillary at the end for self-seeding during the growth process. An in-house Bridgman furnace, which consists of a three-zone tube furnace (EVC 12/450B, 1200 °C, Carbolite Gero) and a vertical motion control system, is used to grow the crystal. The sealed ampoule is suspended in the hot top zone ($T_{max}$ ~ 240 °C) for at least 12 h to ensure that the chemicals melt homogeneously. Then, the ampoule is descended towards the colder bottom zone ($T_{max}$ ~ 80 °C) at a speed of approximately 0.53 mm/h over 14 days. Thereafter, the motor is stopped and the furnace is gradually cooled down to room temperature to prevent thermal shocks. The as-grown crystal is cut using a diamond saw and gently removed from the Pyrex tubing and the inner rod.

### Maser dynamics modelling

To predict and analyze the maser dynamics, we utilize a pair of coupled first-order nonlinear differential equations, namely the Lotka-Volterra model:

$$\dot{N} = -2BNq - \gamma N \qquad (1)$$

$$\dot{q} = -k_c q + BNq \qquad (2)$$

where $N$ and $q$ are the numbers of population inversion and microwave photons in the cavity. $\gamma$ is the spin-lattice relaxation rate and $k_c$ is the photon decay rate in the cavity, which can be calculated from the loaded $Q$ factor as $k_c = \omega/Q_{load}$. $B$ stands for the Einstein coefficient of the stimulated emission, which is directly linked to the resonator's mode volume as follows[32]:

$$B = \frac{\mu_0 \hbar \omega \gamma_e^2 T_2}{4 V_m} \qquad (3)$$

where $\mu_0$ is the vacuum permeability, $\hbar$ the reduced Planck constant, $\gamma_e = 1.76 \times 10^{11}$ rad · s$^{-1}$ · T$^{-1}$ the electron gyromagnetic ratio, and $T_2 = 2.1 \mu s$ the spin-spin relaxation time of pentacene. Spontaneous emission is omitted from our dynamical model because, at microwave frequencies, its rate is vanishingly small compared to stimulated emission.

For the clustered SRR resonating at $\omega = 1.4493$ GHz · $2\pi$, $Q_{load}$ of the maser, including the pc:pt crystal is determined to be 530 from the $S_{11}$ parameter measured using a 2-port vector network analyzer (Rohde & Schwarz, 100 kHz ~ 20 GHz) without laser excitation, resulting in $k_c = 17.2 \times 10^6$ s$^{-1}$. At room temperature $T$, the initial value of $q$ equals the thermal photon number in the cavity without optical excitation, which can be calculated by the Bose-Einstein occupation function $q_0 = (e^{\frac{\hbar \omega}{k_B T}} - 1)^{-1} \sim 4300$, where $k_B$ is the Boltzmann constant. Given proper values for $B$, $\gamma$, and initial value of $N$, the coupled equations can be integrated in time to reveal the maser dynamics. From the measured maser pulse power $P(t)$, the microwave photon number as a function of time is given by $q(t) = P(t)(1 + \kappa)/\hbar \omega k_c \kappa$, where $\kappa = 0.3$ is the coupling coefficient between the magnetic probe and the cavity. Using the differential evolution algorithm[40], the best agreement between the experimental data and the model is achieved with $\gamma = 1.4 \pm 0.4 \times 10^4$ s$^{-1}$, $B = 38.7 \pm 2.9 \times 10^{-8}$ s$^{-1}$ while varying $N_0$ from $4.5 \times 10^{13}$ to $7.3 \times 10^{13}$. We note that the spin-lattice relaxation rate obtained here is slightly higher than the reported value $(1.1 \pm 0.2 \times 10^4$ s$^{-1})$[41]. This difference might stem from the fact that spin decays from the triplet state back to the singlet ground state is not included in the L-V model. Assuming steady-state operation $(\dot{q} = 0)$, the threshold population inversion is given by: $N_{threshold} = k_c/B = 4.4 \times 10^{13}$. Since the Purcell factor scales as $F_p \propto \frac{Q_{load}}{V_m}$, it follows that $N_{threshold} \propto \frac{1}{F_p}$, confirming that an ultrasmall $V_m$ and high $Q_{load}$ (i.e., a large $F_p$) are key to achieving low-threshold, efficient maser operation.

## Data availability

The raw experimental data and the 3D model of the clustered SRR are available on the repository Zenodo under the https://doi.org/10.5281/zenodo.10012323.

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

## Acknowledgements

This work was supported by the Swiss National Science Foundation under the Eccellenza grant No. 181232. We thank Zhe Zhang and Haoye Qin for helpful suggestions in preparing this manuscript.

## Author contributions

R.X. designed the toroidal SRR cluster and performed the numerical simulations under the supervision of R.F. and M.K.M. P.B. designed the vacuum system and performed the sublimation of the p-terphenyl powders. R.X. and P.B. designed the in-house Bridgman furnace and grew the pentacene:p-terphenyl crystals. R.X. fabricated the resonators, conducted the experiments, and performed the data analysis. R.X. and R.F. wrote the manuscript. R.F. supervised the entire project. All authors discussed the results and contributed to the manuscript.

## Competing interests

The authors declare no competing interests.
