## [Transparent Peer Review file · Nature Communications]

All-metallic magnetic Purcell enhancement in a thermally stable room-temperature maser

Corresponding Author: Professor Romain Fleury

Version 0:

Reviewer comments:

Reviewer #1

(Remarks to the Author)

The authors present a novel all-metallic resonator with high Purcell factor that has improved temperature stability compared with the strontium titanate dielectric resonators used in previously published pentacene maser work.

It is well known that strontium titanate has a very high temperature coefficient of permittivity which leads to shifts in the resonator frequency due to temperature changes. The sources of temperature change in the maser could be environmental but would be dominated by the energy deposited in the pentacene-doped gain medium by the optical pump pulses.

The main result of this manuscript is the temperature stability of the maser. However, there is little or no reference to the sensitivity of a the maser to temperature changes, i.e why does the resonant frequency change? Neither the coefficients of thermal expansion of materials, temperature coefficient of permittivity (for the strontium titanate case) nor the temperature coefficient of the zero-field splitting parameters of pentacene are mentioned. Some analysis here would be helpful, by estimating or simulating, by taking the thermal conductivity of the materials into account, how the deposited optical pump energy diffuses through the device. On this note, I noticed that the authors used a polystyrene support material for the strontium titanate dielectric resonator, whereas earlier work used sapphire. These two materials have thermal conductivities that are four orders of magnitude different. Using polystyrene as a support would magnify the effects of the optical pump energy dissipation and so the comparison between strontium titanate and metal may be a little unfair if polystyrene is used.

In any case, wouldn't it be trivial, under periodic optical pump pulses, to adjust the resonator frequency to be resonant with the spin transition once a steady state temperature has been reached?

Although the resonator design is interesting, the work reported here is mostly a reproduction of previous work. For this reason I am afraid that I do not think the result is significant enough for publication in Nature Communications.

Reviewer #2

(Remarks to the Author)

In the work titled "All-metallic magnetic Purcell enhancement in a thermally stable room-temperature maser," the authors present an innovative approach to enhancing the spontaneous and stimulated emission rates of magnetic quantum emitters through the Purcell effect. This study introduces a dielectric-free, all-metallic toroidal split-ring resonator cluster, demonstrating high Purcell factors and maser action at room temperature. The work contains interesting results that could significantly impact the design of high-performance quantum devices. However, before I can recommend it for publication, the work needs revision on the following issues:

1. The main concern is the lack of discussion on why the authors need a large Purcell factor for the generation. The role of the Purcell effect in lasers and masers has been debated for many years. Small mode volumes help with the fast exchange of energy between the active media and the resonator. However, enhancing the Purcell effect also enhances all spontaneous effects, including spontaneous emission, leading to an increase in noise. The authors should discuss this and explain why they need a high Purcell factor. Is this useful to boost the effect further? How does the Purcell factor fit into the Lotka-Volterra model? Without this discussion and analysis, the work remains vague.

2. The authors state in the abstract that "At room temperature, structures constructed using high-index dielectric materials

have been favored due to their low absorption losses,” but this is not true for all spectral ranges. For example, at microwave frequencies, metallic structures are usually more favorable due to their vanishing skin depth.

3. In the abstract, they first say “which are unavoidable during the optical excitation or read-out of the quantum states,” definitely meaning the optical range. However, they then introduce a resonator with a fundamental magnetic mode at 1.45 GHz. There are other examples of this discrepancy. I recommend the authors revise the text to make it more straightforward and concrete.

4. A similar comment applies to terminology like “the magnetic mode.” What is a magnetic mode? A mode of a magnetic field? Likely, the authors mean something different because they know that an oscillating magnetic field cannot exist without a curl electric field. Terms like magnetic mode and magnetic Purcell should be either revised or explained upon introduction. Additionally, what is “extreme” in “such extreme dielectrics” like sapphire?

5. The authors formulate the problem as “as spin transition frequencies are commonly in the gigahertz range, i.e., with emission wavelengths (λ) of the order of tens of centimeters, leveraging the magnetic Purcell effect comes with the inherent challenge of dealing with the large ratio between λ^3 and achievable values of mode volume V_m .” First, the concept of the magnetic Purcell effect appears abruptly here without explanation. Second, and more importantly, the problem is not the ratio of the wavelength to the resonator dimension but the fact that in this range, the mode volume is typically close to λ^3 .

6. The authors make this statement: “Indeed, since the latter represents the effective cavity volume over which the magnetic dipoles are aligned with the magnetic field of the mode, it is drastically limited by our ability to squeeze the magnetic mode into a volume sufficiently small to match the one occupied by the emitters,” without explanation. What do the authors mean by “squeeze the magnetic mode”? Take, for example, a piece of coaxial cable. Coax has no cut-off frequency, and its section can be as small as needed. Would such a coaxial resonator create a strong Purcell effect? Why?

7. I disagree with the statement “At room temperature, small mode volumes and high Q have been achieved by employing dielectrics with very high permittivities” in the context of the importance of the Q-factor. First, metallic high-Q resonators at microwaves are also known. Second, high-index dielectrics not only have high Q-factors but also “squeeze” the wavelength and hence the mode volume.

8. I cannot fully agree with “However, such extreme dielectrics are inherently plagued by their utmost sensitivity to thermal fluctuations, making them excessively prone to detuning.” What thermal fluctuations are the authors talking about? These have different primary sources of noise. The thermal noise in dielectrics is usually due to dipole relaxation processes and phonon interactions. Metals have free electrons that contribute to Johnson-Nyquist noise. I would say that dielectric resonators will typically have lower thermal fluctuations compared to metallic resonators in the microwave range. The authors have to explain what they mean here and probably support their claims with estimations and/or references.

9. The statement “Very small room temperature fluctuations, or the unavoidable thermal heating coming from optical excitation or read-out, would inevitably shift the mode frequency away from the quantum emitter linewidth” is also vague. What regime do the authors mean here, good cavity ($Q_{cav} \gg Q_{emit}$) or bad cavity ($Q_{cav} \ll Q_{emit}$)? If it is a good cavity, which follows from the narrative, then it is less important what happens to the cavity, as any noise from and fluctuations of the emitter will be more important. Could the authors elaborate on this?

10. When the authors make statements about the maser stability like “We demonstrate maser operation stable in frequency and in output pulse shape over thousands of seconds, which is not achievable in prior dielectric-based designs,” they should accompany them with references and examples of the state of the art.

11. The authors should elaborate more on the design of the SRR cluster. Why this geometry? Why does it provide strong confinement of the magnetic energy? Is this number of “almost 350 thousand times smaller than λ^3 ” theoretically limited? If yes, by what means? If not, would it make sense to boost the magnetic field more?

12. Did you take into account the realistic loss and skin depth effects in simulations? Does the squeezing of the magnetic (and I believe, electric) field lead to a proportional increase in losses and Johnson-Nyquist noise?

13. Can the oscillations in Fig.2c be due to other mechanisms beyond the Rabi oscillations? The argument “The Rabi oscillation is also confirmed by the broadened spectrum of the pulse” doesn’t work because any shortening of the pulses in the time domain leads to broadening in the frequency domain. Can this effect, for instance, be caused by collective effects like Dicke or other superradiance?

Reviewer #3

(Remarks to the Author)

Review of NCOMMS-23-52768

The paper describes a specially designed metallic resonator employed in conjunction with a pentacene-based optically pumped room temperature pulsed maser to facilitate frequency-stable operation with minimal heating effects. The authors show that such heating effects are detrimental in a similar device based on a dielectric resonator made of SrTiO₃ single crystal, leading to frequency drifts and the eventual loss of maser signal after approximately 1-2 hours. The main novelty claim of this paper is the use of a unique 3D metallic split-ring resonator with a small mode volume and a relatively large quality factor. This results in a resonator with a relatively high Purcell factor (which is approximately $\frac{\lambda^3}{Q \cdot \text{mode volume}}$) that is imperative for facilitating the operation of the maser device.

I found this paper interesting and informative, but I fail to see significant novelty or impact. My main concern is that the use of small metallic volume resonators is known in the field of electron spin resonance, which seems to undermine the main novel point of the paper. Recent examples of resonators with similar or even smaller mode volumes (in units of λ^3) can be found here: [arXiv:2307.11269](https://arxiv.org/abs/2307.11269) and references therein. Additionally, there is the possibility of using metallic loop-gap resonators of similar or smaller size in conjunction with highly temperature-stable dielectric materials, as discussed in the following references:

- [NCBI](https://www.ncbi.nlm.nih.gov/pmc/articles/PMC6948142/)
- [Patent](https://patentimages.storage.googleapis.com/77/0b/52/097e05c3bd3d96/WO2013175235A1.pdf)
- [Wiley](https://onlinelibrary.wiley.com/doi/abs/10.1002/mrm.28595)

While these designs do not necessarily have the exact properties of the resonator presented in the current work, they achieve similar high Purcell factors and use either all-metallic or dielectric materials with negligible temperature dependence of permittivity.

Moreover, comparing the temperature stability of metallic structures to SrTiO₃ seems unfair, as the latter has permittivity drifts in the range of approximately 1% per degree. The same type of maser was realized in the past with a sapphire-based structure that is far more temperature stable.

Another major concern is that the paper discusses the use of a pulsed maser device with very limited practical use, if at all. What would happen if one wishes to use a similar resonator for continuous-wave operation, such as a maser based on NV centers in diamond? At some point, the metal itself may heat up and expand.

Lastly, even with SrTiO₃, the authors show that the drift is very slow. Is it worth having a worse resonator design (with a lower Purcell factor than SrTiO₃) but better temperature stability? Possibly, it would be better to automatically tune back the SrTiO₃ resonator by changing the metallic cup on it. What I am trying to say is that the fundamental problem (heating) which the authors claim to exist may have other solutions that would keep the original design with the high Purcell factor. And on a side note, I am not sure this problem is real if one uses more temperature-stable dielectrics.

Other minor remarks:

1. The Introduction discusses the Purcell enhancement factor in relation to increasing the spontaneous emission rate. However, the experimental results do not show any change in this rate compared to the usual T₁ of the sample (as shown, for example, [here](https://www.nature.com/articles/nature16944)). In my opinion, it would have been better to describe the requirements for achieving maser action through the cooperativity factor. The discussion of changing the spontaneous emission rate seems out of place. Indeed, the Purcell factor is part of the cooperativity factor, but this should be taken in the context of achieving maser threshold (having more gain than losses inside the cavity) and not connected to the spontaneous emission rate.
2. Page 2 – the expected frequency variations of SrTiO₃-based resonators: can you be more quantitative? For SrTiO₃, it is clear that there would be relatively large changes, but for sapphire, this is a very small change in permittivity. What about the expected change in resonator frequency due to metal expansion?
3. Mode volume should refer to the magnetic field in all directions (namely magnitude) in the denominator since it relates to total magnetic energy. What is $\langle H_z^2 \rangle(r)$ and how is it dependent on r ?
4. Figure 1 – can you plot the E-field as well?
5. Page 4 – it should be mentioned clearly that this maser operates only in pulsed mode. What about masers that need a static field? The all-metallic design with its shield is larger than the dielectric design, implying a larger and bulkier magnet.
6. Page 8 – what is the spin concentration of the sample?

Version 1:

Reviewer comments:

Reviewer #1

(Remarks to the Author)

Second review

R 1.1

The authors state that they do not observe any frequency shift using the all-metallic resonator. I find this comment confusing since they state later in the rebuttal they estimate the temperature coefficient of frequency to be 130 ppm/K. The shift here being due to thermal expansion of the silver. This is substantial and should be measurable at the frequency of the maser emission at 1.45 GHz.

R 1.2

The authors have produced graphs showing temperature changes in the resonator. No detail on how these graphs were generated is given so I can not comment further except to say that they authors did not properly address the original concerns.

R 1.3

Whilst an improved resonator temperature stability is admirable, I remain unconvinced that the work presented here is significant enough a development to warrant publication in Nature Communications. As pointed out by Reviewer 2, a patent was published in 2013 (WO 2013/175235 A1, see Figure 4) that describes using a split-ring resonator, very similar to that reported by the authors, suggesting that the idea is not original.

Reviewer #2

(Remarks to the Author)

The authors have addressed all my comments, and I now recommend the paper for publication.

Reviewer #3

(Remarks to the Author)

Review of NCOMMS-23-52768

The authors have made efforts to address the reviewers' comments and revise their manuscript accordingly. I see improvements; however, there are still several issues that I believe require further attention:

1. (R3.3) As noted in my previous report, this work focuses on pulsed maser operation at a very low repetition rate (1 Hz), which has limited applicability. The authors provided in their response some estimates regarding the feasibility of supporting continuous-wave (CW) diamond NV-based masers—especially in relation to the resonator—but these were not incorporated into the revised paper.
 2. (R2.1) The discussion of the spontaneous emission rate and the Purcell factor is non-quantitative. Additionally, the relationship to the Einstein B-coefficient does not seem to be incorporated into the revised paper.
 3. (R2.4) The definition of the mode volume used in the text is somewhat controversial in this context. Specifically, normalizing by the maximum of the magnetic field can produce artificially low “magnetic mode” volumes if there is a localized magnetic field “hot spot.” In contrast, if the microwave magnetic field is relatively homogeneous across the resonator volume, the chosen definition might be reasonable. However, a more relevant approach could be to define the 50% or 90% mode volume, referring to the region where 50% or 90% of the microwave magnetic energy is localized. This might explain the discrepancy between the measured data and the mode volume calculation (a factor of ~3). I believe the authors should discuss this point in more detail.
 4. (R2.5a) The size of the crystal and the specific dimensions of the device are crucial details that should be linked more explicitly to the required optimal resonator geometry. This appears to be another instance where the authors' response did not result in changes to the manuscript.
 5. (R2.8) Here as well, I believe the text of the paper should be modified accordingly, reflecting the issues raised in the response.
 6. (R2.9) It seems somewhat unusual to discuss frequency stability using MHz-scale data, presumably due to the short-pulse operation. Comparing this to the ~10 kHz linewidths of CW maser systems cited in the references is difficult. A more direct comparison or acknowledgment of the differences in operational modes would be helpful.
 7. (R3.1) The “quasi-2D” structures mentioned by the authors in relation to the quoted references of previous work have heights of approximately 0.5 mm and 1.4 mm—borderline for the intended application. However, there is no fundamental reason why these could not be extended to heights of, for example, 3 mm or more to fit the crystal used in this work. It is therefore unclear why the authors dismiss these structures as unsuitable for the current application.
 8. (R3.3) The condition $C > 1$ is typically considered indicative of strong (rather than weak) coupling.
- Overall, while the revised manuscript addresses some of the reviewers' concerns, I believe further clarifications and revisions are needed in the areas detailed above.

Response letter, manuscript number NCOMMS-23-52768

We would like to thank all the reviewers for their time and careful review of our manuscript. In the following, we provide a point-by-point response to each of the Reviewer's comments.

Reviewer 1:

R1.0: The authors present a novel all-metallic resonator with high Purcell factor that has improved temperature stability compared with the strontium titanate dielectric resonators used in previously published pentacene maser work.

Response: We thank the reviewer for his/her careful review and very useful comments on our work, that greatly helped us to improve it and clarify some important points.

R1.1: It is well known that strontium titanate has a very high temperature coefficient of permittivity which leads to shifts in the resonator frequency due to temperature changes. The sources of temperature change in the maser could be environmental but would be dominated by the energy deposited in the pentacene-doped gain medium by the optical pump pulses.

Response: We totally agree with the reviewer that thermal heating, caused by consecutive optical excitation, could happen in the resonator and the pentacene doped gain medium. In our study, the masers constructed using the all-metallic resonator and the SrTiO₃ resonator were tested under the same conditions (e.g., optical excitation path, laser pulse energy and repetition rate, etc.), and therefore the heat deposited in the gain media is comparable. However, we don't observe any frequency shift with the all-metallic resonator, which indicates that the thermal effects that shift the frequency of the X ↔ Z transition are negligible, and the maser constructed using the proposed resonator is indeed more thermally robust.

R1.2: The main result of this manuscript is the temperature stability of the maser. However, there is little or no reference to the sensitivity of the maser to temperature changes, i.e why does the resonant frequency change? Neither the coefficients of thermal expansion of materials, temperature coefficient of permittivity (for the strontium titanate case) nor the temperature coefficient of the zero-field splitting parameters of pentacene are mentioned. Some analysis here would be helpful, by estimating or simulating, by taking the thermal conductivity of the materials into account, how the deposited optical pump energy diffuses through the device. On this note, I noticed that the authors used a polystyrene support material for the strontium titanate dielectric resonator, whereas earlier work used sapphire. These two materials have thermal conductivities that are four orders of magnitude different.

Using polystyrene as a support would magnify the effects of the optical pump energy dissipation and so the comparison between strontium titanate and metal may be a little unfair if polystyrene is used.

Response: We thank the reviewer for all the relevant remarks that have helped us improve the analysis and presentation of our findings. We are happy to provide quantitative arguments regarding thermal sensitivity. The resonant frequency in the SrTiO₃-based maser changed mainly due to the high temperature coefficient of the permittivity of SrTiO₃, which leads to a large temperature coefficient of frequency of approximately 1700 ppm/K^{1,2}.

As for the all-metallic SRR cluster, one main potential reason of frequency drifting would be thermal expansion due to thermal heating by the laser. Based on the volumetric thermal expansion coefficient of silver around $54 \times 10^{-6} \text{ K}^{-1}$, the temperature coefficient of resonant frequency of the all-metallic SRR cluster is estimated to be approximately 130 ppm/K.

The polystyrene support was utilized for both the all-metallic resonator and the SrTiO₃ resonator, but for different purposes. For the SrTiO₃ resonator, the polystyrene support was employed to reduce ohmic losses on the bottom copper plate, thereby improving the Q factor. In the case of the all-metallic resonator, the polystyrene support was used solely to elevate the resonator for optimal positioning of the magnetic probe. Without this support, the side hole on the copper cylinder—used to insert the probe—would be too high for proper alignment with the all-metallic resonator. We apologize for the lack of clarity on this point in the original manuscript. This information is now included in Supplementary Fig.S3 of the revised manuscript.

To study how the deposited optical pump energy diffuses through the SrTiO₃ resonator with different support materials, we conducted simulations in COMSOL. In the experiment, a laser pulse delivered via an optical fiber had a pulse duration of 5 ns, a repetition rate of 1 Hz, and a pulse energy of 3 mJ; the same parameters were applied in the simulation. As shown in Video R1.2.1-T vs time-sapphire support and Video R1.2.2-T vs time-airfoam support, the laser was active for the first 5 ns of each second, heating the SrTiO₃ resonator. The heat dissipated was concentrated in the region near the optical fiber's end. After the laser pulse ended, the accumulated heat gradually dissipated.

Figure R1.2.1 illustrates the variation in maximum and average temperature of the SrTiO₃ resonator over five seconds. The results demonstrate that the choice of support material did not result in significant differences in the temperature variation of the SrTiO₃ resonator. While these materials may cause some differences over longer experimental durations, the primary factor contributing to the frequency shift of the SrTiO₃ resonator remains its poor temperature coefficient of resonant frequency.

Figure R1.2.1 Maximum (a) and average (b) temperature variation of the SrTiO₃ resonator placed on sapphire support (blue solid line) and polystyrene foam support (red dotted line). The pump laser had a repetition rate of 1Hz, pulse duration 5ns and pulse energy of 3 mJ.

Revision: We have added the following text in the Results section:

- This excellent thermal stability is attributed to silver's high thermal conductivity and moderate thermal expansion coefficient, resulting in a temperature coefficient of resonant frequency of approximately 130 ppm/K. In contrast, the central frequency of the SrTiO₃ based maser shifts from 1.449 GHz to 1.453 GHz, resulting from SrTiO₃'s large temperature coefficient of resonant frequency (approximately 1700 ppm/K). We simulated the diffusion of the deposited optical pump energy and the resulting temperature variations in the SrTiO₃ resonator using COMSOL. The simulation results are presented in the Supplementary Movie1 and Fig.S6.

We have added Movie1 and Fig.S6 in the Supplementary materials.

R1.3: In any case, wouldn't it be trivial, under periodic optical pump pulses, to adjust the resonator frequency to be resonant with the spin transition once a steady state temperature has been reached?

Although the resonator design is interesting, the work reported here is mostly a reproduction of previous work. For this reason, I am afraid that I do not think the result is significant enough for publication in Nature Communications.

Response: The resonant frequency of the SrTiO₃ resonator can be manually tuned using the movable top copper plate. However, achieving and maintaining resonance with precision would require frequent adjustments and physical access to the copper cavity, making this process maintenance-intensive and potentially impractical for long-term experiments. Alternatively, while automatic retuning with a feedback loop could improve accuracy, it would add substantial complexity to the system, impacting simplicity and ease of use. Furthermore, stable operation without frequency drift is critical for certain applications, such as weak magnetic field sensing³.

In addition, it is worth noting that the thermal conductivity (κ) of metals, such as silver or copper, is more than an order of magnitude higher than that of dielectric materials like SrTiO₃ and Al₂O₃.

According to Fourier's law, the temperature increase ΔT of a material is inversely proportional to κ . This implies that, under comparable heat flux conditions, the steady state temperature of dielectric resonators would be higher than that of metallic ones of similar sizes and would be more susceptible to thermal fluctuations.

While we recognize the foundational contributions of previous work, our study offers significant advancement through the introduction of a fully metallic structure. This unique design achieves exceptional thermal stability and robust three-dimensional magnetic energy confinement at a deep subwavelength scale, resulting in an efficient and maintenance-free platform.

Reviewer 2:

R2.0: In the work titled "All-metallic magnetic Purcell enhancement in a thermally stable room-temperature maser," the authors present an innovative approach to enhancing the spontaneous and stimulated emission rates of magnetic quantum emitters through the Purcell effect. This study introduces a dielectric-free, all-metallic toroidal split-ring resonator cluster, demonstrating high Purcell factors and maser action at room temperature. The work contains interesting results that could significantly impact the design of high-performance quantum devices. However, before I can recommend it for publication, the work needs revision on the following issues.

Response: We thank the reviewer for his/her careful review and very useful comments on our work, that greatly helped us to improve it and clarify some important points.

R2.1: The main concern is the lack of discussion on why the authors need a large Purcell factor for the generation. The role of the Purcell effect in lasers and masers has been debated for many years. Small mode volumes help with the fast exchange of energy between the active media and the resonator. However, enhancing the Purcell effect also enhances all spontaneous effects, including spontaneous emission, leading to an increase in noise. The authors should discuss this and explain why they need a high Purcell factor. Is this useful to boost the effect further? How does the Purcell factor fit into the Lotka-Volterra model? Without this discussion and analysis, the work remains vague.

Response: A high Purcell factor plays a pivotal role in achieving robust and efficient maser action at room temperature. By enhancing the interaction between the active medium and the cavity mode, a large Purcell factor increases the stimulated emission rate and effectively reduces the pumping threshold for maser operation⁴. While this does raise spontaneous emission and its associated noise, improved efficiency, reduced threshold, and device compactness often outweigh the drawbacks.

Regarding the Lotka-Volterra model, the Purcell factor indeed enters it directly. The mode volume V_m , which is relevant to the Purcell factor, can be related to the Einstein coefficient B as follows⁵:

$$B = \frac{\mu_0 \hbar \omega \gamma_e^2 T_2}{4V_m}$$

Where μ_0 is the permeability of free space, \hbar is the reduced Planck constant, ω is the maser angular frequency, γ_e is the Gyromagnetic ratio of electrons and T_2 is the spin-spin relaxation time.

R2.2: The authors state in the abstract that “At room temperature, structures constructed using high-index dielectric materials have been favored due to their low absorption losses,” but this is not true for all spectral ranges. For example, at microwave frequencies, metallic structures are usually more favorable due to their vanishing skin depth.

Response: We thank the reviewer for this clarification, which we agree with. We will revise our manuscript to provide a more accurate and nuanced perspective.

Revision: We have revised the sentence in the abstract from:

- “At room temperature, structures constructed using high-index dielectric materials have been favored due to their low absorption losses.”

to:

- “At room temperature, structures constructed using high-index dielectric materials have been favored due to their ability to effectively confine electromagnetic fields.”

R2.3: In the abstract, they first say “which are unavoidable during the optical excitation or read-out of the quantum states,” definitely meaning the optical range. However, they then introduce a resonator with a fundamental magnetic mode at 1.45 GHz. There are other examples of this discrepancy. I recommend the authors revise the text to make it more straightforward and concrete.

Response: We appreciate the reviewer’s observation and would like to clarify this point, which is not a discrepancy. While the resonator itself operates at a microwave frequency (1.45 GHz), the underlying quantum systems involved are often prepared, manipulated, and read out optically. For example, in pentacene-doped p-terphenyl crystals and NV centers in diamond, optical pumping is necessary for achieving population inversion and subsequently generating microwave emission. Additionally, many quantum sensing protocols rely on optical excitation or detection of states whose coherent transitions occur in the microwave regime⁶.

When we mention “unavoidable thermal fluctuations during the optical excitation or read-out of the quantum states,” we are referring to the practical challenges introduced by these optical steps. Although the masing frequency is in the microwave domain, the necessary optical processes can still introduce thermal loads and fluctuations that affect the system’s overall stability and noise performance.

R2.4: A similar comment applies to terminology like “the magnetic mode.” What is a magnetic mode? A mode of a magnetic field? Likely, the authors mean something different because they know that an oscillating magnetic field cannot exist without a curl electric field. Terms like magnetic mode and magnetic Purcell should be either revised or explained upon introduction. Additionally, what is “extreme” in “such extreme dielectrics” like sapphire?

Response: We apologize for the lack of clarity in our terminology, which we are happy to clarify. What we call a magnetic mode in the context of our masers is an electromagnetic mode whose magnetic energy density strongly dominates the electrical energy density over the volume of the gain medium (although as for any electromagnetic eigenmode, the total electric and magnetic energies, integrated over space, are equal). As our goal is to enhance the emission rate of magnetic dipole emitters, we have chosen a resonator mode configuration that maximizes the local magnetic field component and its alignment to the magnetic quantum emitters. In this context, we refer to it as a “magnetic mode” to emphasize the strong magnetic field concentrated in the gain region and aligned with the emitters.

Regarding the phrase “extreme dielectrics,” our intention was to describe materials like SrTiO₃, which have exceptionally high dielectric constants ($\epsilon_r \approx 320$). To avoid confusion, we will rephrase “extreme dielectrics” as “dielectrics with exceptionally high dielectric constants” in the revised manuscript.

Revision: We have revised the sentence in the introduction from:

- “However, such extreme dielectrics are inherently plagued by their utmost sensitivity to thermal fluctuations, making them excessively prone to detuning.”

to:

- “However, dielectrics with extremely high permittivities are inherently sensitive to thermal fluctuations, making them excessively prone to detuning.”

R2.5a: The authors formulate the problem as “as spin transition frequencies are commonly in the gigahertz range, i.e., with emission wavelengths (λ m) of the order of tens of centimeters, leveraging the magnetic Purcell effect comes with the inherent challenge of dealing with the large ratio between λ^3 and achievable values of mode volume V_m .” First, the concept of the magnetic Purcell effect appears abruptly here without explanation. Second, and more importantly, the problem is not the ratio of the wavelength to the resonator dimension but the fact that in this range, the mode volume is typically close to λ^3 . The authors make this statement: “Indeed, since the latter represents the effective cavity volume over which the magnetic dipoles are aligned with the magnetic field of the mode, it is drastically limited by our ability to squeeze the magnetic mode into a volume sufficiently small to match the one occupied by the emitters,” without explanation. What do the authors mean by “squeeze the magnetic mode”?

Response: The magnetic Purcell effect refers to the enhancement of the spontaneous emission rate of magnetic dipole transitions when they couple to the magnetic field of a resonant mode. At microwave frequencies—where spin transitions often reside—the wavelength is much larger compared to the typical size of the gain media that can practically be grown (whether they are organic crystals or diamond-based). The mode volumes that can practically be achieved with standard metallic cavity resonators typically remain close to λ^3 , and therefore are too big, and the ones

achieved with high index dielectrics are thermally sensitive. The point we make is precisely that we managed to create an all-metallic resonator that is not too big and provides a field that overlaps well with the emitters.

By “squeezing the magnetic mode,” we mean engineering the resonator geometry and boundaries to reduce the effective volume over which the magnetic field is both significant and well-aligned with the emitter’s magnetic dipole moment. The goal is to confine the field as tightly as possible around the emitters to maximize coupling and hence increase their emission rate.

R2.5b: Take, for example, a piece of coaxial cable. Coax has no cut-off frequency, and its section can be as small as needed. Would such a coaxial resonator create a strong Purcell effect? Why?

Response: While a coaxial resonator can have a small cross-sectional area, it still needs a length on the order of $\lambda/2$ to make a resonance, and the magnetic field is azimuthal. We would therefore not only need a gain medium with azimuthally oriented magnetic dipoles (which seems extremely challenging in a monocrystal since the emitters align along a definite crystal direction during crystal growth), but also this gain medium would need to be at least $\lambda/2$ long. Therefore, this example does not evade our argument that making an overlap between a resonant magnetic field and the gain media is challenging with all-metallic structures.

R2.7: I disagree with the statement “At room temperature, small mode volumes and high Q have been achieved by employing dielectrics with very high permittivities” in the context of the importance of the Q-factor. First, metallic high-Q resonators at microwaves are also known. Second, high-index dielectrics not only have high Q-factors but also “squeeze” the wavelength and hence the mode volume.

Response: We agree that metallic high-Q microwave resonators are known, but they have never been successfully employed to make a MASER. Our work is the first to achieve this. This is why they are not listed in the state of the art about room-temperature maser resonators.

The second part of the referee’s comment (“Second, ...”), we completely agree with. These dielectric resonators have been used, they indeed also squeeze the wavelength and the mode volume. Our point is that they come with the drawback of thermal sensitivity (which is our next sentence).

Revision: We have revised the text and made sure not to make any claim that can be misunderstood.

R2.8: I cannot fully agree with “However, such extreme dielectrics are inherently plagued by their utmost sensitivity to thermal fluctuations, making them excessively prone to detuning.” What thermal fluctuations are the authors talking about? These have different primary sources of noise. The thermal noise in dielectrics is usually due to dipole relaxation

processes and phonon interactions. Metals have free electrons that contribute to Johnson-Nyquist noise. I would say that dielectric resonators will typically have lower thermal fluctuations compared to metallic resonators in the microwave range. The authors have to explain what they mean here and probably support their claims with estimations and/or references. The statement “Very small room temperature fluctuations, or the unavoidable thermal heating coming from optical excitation or read-out, would inevitably shift the mode frequency away from the quantum emitter linewidth” is also vague. What regime do the authors mean here, good cavity ($Q_{\text{cav}} \gg Q_{\text{emit}}$) or bad cavity ($Q_{\text{cav}} \ll Q_{\text{emit}}$)? If it is a good cavity, which follows from the narrative, then it is less important what happens to the cavity, as any noise from and fluctuations of the emitter will be more important. Could the authors elaborate on this?

Response: We apologize for the use of the word “fluctuations” which may have misled the reviewer into thinking that we were discussing noise-related effects. We are referring to temperature variations within the dielectric resonator material. These variations can be caused by environmental temperature changes or local heating from processes such as optical pumping. In high-permittivity dielectric materials, such as SrTiO₃, the temperature coefficient of the permittivity is normally high. Therefore, even small temperature changes can lead to noticeable shifts in the resonance frequency, effectively “detuning” the resonator.

This concern is distinct from the fundamental thermal noise mechanisms in dielectrics (such as dipole relaxation or phonon interactions) or the Johnson-Nyquist noise present in metals. Rather, our point is that highly temperature-sensitive resonators may require more stringent thermal stabilization to maintain the desired resonance conditions.

R2.9: When the authors make statements about the maser stability like “We demonstrate maser operation stable in frequency and in output pulse shape over thousands of seconds, which is not achievable in prior dielectric-based designs,” they should accompany them with references and examples of the state of the art.

Response: The examples of frequency fluctuation due to laser heating during operation can be found in papers:

1. Breeze, J., Salvadori, E., Sathian, J. et al. Continuous-wave room-temperature diamond maser. Nature 555, 493–496 (2018)
2. Salvadori, E., Breeze, J., Tan, KJ. et al. Nanosecond time-resolved characterization of a pentacene-based room-temperature MASER. Sci Rep 7, 41836 (2017).

Revision: We have added these references in the same text in the revised manuscript.

R2.10.a: The authors should elaborate more on the design of the SRR cluster. Why this geometry? Why does it provide strong confinement of the magnetic energy?

Response: The chosen geometry is inspired by metamaterial unit cells—split-ring resonators (SRRs) — which inherently support sub-wavelength magnetic resonances. We have added a detailed explanation and design steps in the Supplementary Materials (Supplementary Fig. S7).

Revision: The detailed design procedure of the SRR cluster has been added in the Supplementary Materials (Supplementary Fig. S7).

R2.10.b: Is this number of “almost 350 thousand times smaller than λ^3 ” theoretically limited? If yes, by what means? If not, would it make sense to boost the magnetic field more?

Response: We do not believe that this number is theoretically limited, however within the constraints of operating with silver and the current fabrication methods, it is the best we could achieve.

By arranging multiple SRRs in close proximity, their interactions produce electromagnetic modes that concentrate the field into volumes significantly smaller than λ^3 . The specific figure of “350 thousand times smaller than λ^3 ” results from an optimization process aimed at striking an effective balance among the physical size of the resonator, the magnetic mode volume, and the quality factor. Pushing the mode volume too aggressively below this optimized point typically increases absorption losses, lowering the Q too drastically. In practice, the design is therefore limited by absorption losses, which occur at the surface of the metal and are strongly dependent on the surface roughness. We employed the more advanced casting techniques to make the best possible prototype with silver: optimization may require new casting methods with less deformation of the geometry upon cooling and better roughness.

With materials with lower losses, it is possible to beat this limit: a known example is superconducting circuits. There is thus no theoretical bound, but if we do not allow cryogenic temperatures, with silver we are close to the optimum.

R2.12: Did you take into account the realistic loss and skin depth effects in simulations? Does the squeezing of the magnetic (and I believe, electric) field lead to a proportional increase in losses and Johnson-Nyquist noise?

Response: Yes, our simulations incorporate the finite conductivity and material parameters of both metallic and dielectric components, allowing us to account for ohmic losses and the associated skin depth limitations. As the electromagnetic field is confined into a smaller volume, its local energy density increases. This, in principle, can lead to higher ohmic losses and an increase in Johnson-Nyquist noise, as both are related to the material’s response to the intensified fields. Our optimization process was therefore not only focused on minimizing mode volume but also on managing these losses to achieve a practical compromise between field confinement and Q factor.

R2.13: Can the oscillations in Fig.2c be due to other mechanisms beyond the Rabi oscillations? The argument “The Rabi oscillation is also confirmed by the broadened spectrum of the pulse” doesn’t work because any shortening of the pulses in the time domain leads to broadening in the frequency domain. Can this effect, for instance, be caused by collective effects like Dicke or other superradiance?

Response: Yes, the reviewer is right.

Revision:

1. We have revised the caption of Fig.2 from

- With stronger optical pumping energy, the maser pulse possesses a weak Rabi-oscillatory behavior which is also reflected in its broadened spectrum.

to

- With higher pump energy, the maser pulse exhibits a weak oscillatory behavior and corresponding spectral broadening.

2. We have revised the following text in the Results section from:

- With pumping energy well above the threshold ($N \sim 6.8 \times 10^{13}$), we observe a weak Rabi-oscillatory behavior in the maser pulse, resulting from the coherent energy exchange between the spin ensembles in pentacene and the microwave photons in the cavity. The Rabi oscillation is also confirmed by the broadened spectrum of the pulse caused by the normal mode splitting (Fig.2.c, bottom).

to

- With pumping energy well above the threshold ($N \sim 6.8 \times 10^{13}$), we observe a weak oscillatory behavior in the maser pulse. This behavior could potentially arise from coherent energy exchange between the spin ensemble and the cavity mode, or from collective emission phenomena such as superradiance. Further experimental and theoretical investigations will be necessary to determine the precise underlying mechanism of these oscillations.

Reviewer 3:

R3.0: The paper describes a specially designed metallic resonator employed in conjunction with a pentacene-based optically pumped room temperature pulsed maser to facilitate frequency-stable operation with minimal heating effects. The authors show that such heating effects are detrimental in a similar device based on a dielectric resonator made of SrTiO₃ single crystal, leading to frequency drifts and the eventual loss of maser signal after approximately 1-2 hours. The main novelty claim of this paper is the use of a unique 3D metallic split-ring resonator with a small mode volume and a relatively large quality factor. This results in a resonator with a relatively high Purcell factor (which is approximately $\frac{\lambda^3 \cdot Q}{\text{mode volume}}$) that is imperative for facilitating the operation of the maser device.

Response: We appreciate the Referee's constructive and insightful comments, which have prompted significant revisions that have substantially strengthened our work.

R3.1: I found this paper interesting and informative, but I fail to see significant novelty or impact. My main concern is that the use of small metallic volume resonators is known in the field of electron spin resonance, which seems to undermine the main novel point of the paper. Recent examples of resonators with similar or even smaller mode volumes (in units of λ^3) can be found here: [arXiv:2307.11269] (<https://arxiv.org/abs/2307.11269>) and references therein. Additionally, there is the possibility of using metallic loop-gap resonators of similar or smaller size in conjunction with highly temperature-stable dielectric materials, as discussed in the following references:

- [NCBI](<https://www.ncbi.nlm.nih.gov/pmc/articles/PMC6948142/>)

-[Patent]

(<https://patentimages.storage.googleapis.com/77/0b/52/097e05c3bd3d96/WO2013175235A1.pdf>)

- [Wiley](<https://onlinelibrary.wiley.com/doi/abs/10.1002/mrm.28595>)

While these designs do not necessarily have the exact properties of the resonator presented in the current work, they achieve similar high Purcell factors and use either all-metallic or dielectric materials with negligible temperature dependence of permittivity.

Response: We thank the reviewer for providing these references, which have broadened our perspective, and we have added them in the revised manuscript. We would like to address our work's novelty in two key aspects: three-dimensional magnetic energy confinement and all-metallic strong magnetic Purcell enhancement at subwavelength scales.

While small metallic volume resonators are well-studied in electron spin resonance, the first three cited references (arXiv:2307.11269, NCBI, and Patent) describe quasi-2D structures. However, for applications requiring three-dimensional samples or efficient volumetric optical excitation, achieving energy confinement in three dimensions is essential. Our all-metallic resonator fulfills this need, providing robust 3D magnetic energy confinement with subwavelength compactness and thermal stability, without additional dielectric materials that can complicate system design.

In the last reference [Wiley], the 3D loop-gap ring resonator made from copper, has a simulated mode volume around 0.154 cm^3 and a measured Q factor around 60 at $f_r = 2.3 \text{ GHz}$. These parameters yield a Purcell factor approximately 6.5×10^4 , which is considerably lower than that achieved by our proposed resonator — 77 times lower than our experimental value and 290 times lower than our simulated value.

Revision: The references are added.

R3.2: Moreover, comparing the temperature stability of metallic structures to SrTiO3 seems unfair, as the latter has permittivity drifts in the range of approximately 1% per degree. The same type of maser was realized in the past with a sapphire-based structure that is far more temperature stable.

Response: We compared the temperature stability of our proposed all-metallic structure to the SrTiO3 resonator due to their similar sub-wavelength size and comparable magnetic Purcell factor. The sapphire-based structure is indeed more stable thermally but has a much lower dielectric constant, leading to a device around 200 times larger in size, and almost 10 times lower Purcell factor. This requires a higher optical pumping threshold⁴. In contrast, our proposed all-metallic structure is thermally stable while maintaining compact size and high Purcell factor.

R3.3: Another major concern is that the paper discusses the use of a pulsed maser device with very limited practical use, if at all. What would happen if one wishes to use a similar resonator for continuous-wave operation, such as a maser based on NV centers in diamond? At some point, the metal itself may heat up and expand.

Response: In a maser employing NV centers in diamond with a sapphire resonator, a continuous laser pump power of 400 mW resulted in a frequency shift of 25 MHz, attributed to an estimated $\Delta T_{Al_2O_3} = 35^\circ\text{C}$ temperature increase in both the sapphire resonator and diamond caused by laser heating⁷. Assuming the toroidal SRR cluster has similar size and is operated under the same conditions as the sapphire resonator, the temperature increase ΔT_{SRR} could be estimated as $\Delta T_{SRR}/\Delta T_{Al_2O_3} \sim \kappa_{Al_2O_3}/\kappa_{Ag} \approx 0.1$, following the Fourier's law, where $\kappa_{Al_2O_3}$ and κ_{Ag} are the thermal conductivities of sapphire and silver, respectively. Given that the volumetric thermal expansion coefficient of silver is approximately $54 \times 10^{-6} \text{ K}^{-1}$, the geometric expansion would be less than 0.02%. Consequently, the frequency variation in the SRR cluster is expected to be negligible

compared to that observed in the sapphire resonator. We believe that our resonator would still be relevant in these experiments.

R3.4: Lastly, even with SrTiO₃, the authors show that the drift is very slow. Is it worth having a worse resonator design (with a lower Purcell factor than SrTiO₃) but better temperature stability? Possibly, it would be better to automatically tune back the SrTiO₃ resonator by changing the metallic cup on it. What I am trying to say is that the fundamental problem (heating) which the authors claim to exist may have other solutions that would keep the original design with the high Purcell factor. And on a side note, I am not sure this problem is real if one uses more temperature-stable dielectrics.

Response: During the review of the manuscript, we have had discussions with several authors of studies using room-temperature masers, including at the PhD defense of the first author of this manuscript, who confirmed that this is an important problem. Some colleagues asked us for our resonators and reproduced our experiment. Others came to our lab to see the maser in action. Therefore, we are confident that our study has value.

We agree that the drift in SrTiO₃ is relatively slow, however, stable operation with robust frequency fidelity is essential for certain applications, such as sensors for weak magnetic fields³. In many contexts, stable and repeated pulse operation is favored over continuous operation, because it uses much less input power. Our all-metallic resonator, though currently exhibiting a somewhat lower Purcell factor than theoretical expectations due to fabrication limitations, provides intrinsic thermal stability that ensures reliable, long-term performance without requiring ongoing adjustments. We anticipate that advancements in fabrication techniques could further enhance the Purcell factor in future iterations of this design.

We acknowledge that adaptive tuning methods, such as adjusting the metallic top plate, could potentially address thermal drift in the SrTiO₃ resonator. However, they are challenging to implement, and as a matter of fact, they have not been demonstrated. On the other hand, temperature-stable dielectrics, such as Barium Magnesium Tantalate, could also be considered; however, they typically result in a larger resonator size and, consequently, a greater volume due to lower dielectric constant.

We would like to emphasize that our aim is to develop a platform that combines thermal stability with subwavelength compactness and strong Purcell enhancement, comparable to that achieved by ultra-high-dielectric-constant materials like strontium titanate. Our all-metallic resonator thus offers an optimal blend of these qualities in a simple, efficient, and maintenance-free design.

R3.5: Other minor remarks:

- 1. The Introduction discusses the Purcell enhancement factor in relation to increasing the spontaneous emission rate. However, the experimental results do not show any change in this rate compared to the usual T₁ of the sample (as shown, for example, [here])**

(<https://www.nature.com/articles/nature16944>)). In my opinion, it would have been better to describe the requirements for achieving maser action through the cooperativity factor. The discussion of changing the spontaneous emission rate seems out of place. Indeed, the Purcell factor is part of the cooperativity factor, but this should be taken in the context of achieving maser threshold (having more gain than losses inside the cavity) and not connected to the spontaneous emission rate.

Response: We thank the reviewer for this insightful remark. A discussion on cooperativity has been included in the revised manuscript. As a matter of fact, other experts of the field contacted us during the review process with the request to also report cooperativity.

Revision: In the discussion section of the revised manuscript, we have added:

- “Furthermore, to assess the interaction strength between the emitters and the confined EM mode in the resonator, the cooperativity of the maser, constructed with the SRR cluster, can be calculated using $C = 4g^2N/k_c k_s$, where $k_s = 2/T_2^*$ is the spin dephasing rate with the spin dephasing time $T_2^* \approx 2.9 \mu\text{s}$ and $g = \gamma_e \sqrt{\mu_0 \hbar \omega / 2V_m}$ is the single spin-photon coupling strength. With $N = \max(N_0) = 7.3 \times 10^{13}$, the cooperativity is estimated to be $C \approx 6.4$, suggesting a relatively weak coupling between the emitters and the SRR cluster”.

R3.6: 2. Page 2 – the expected frequency variations of SrTiO₃-based resonators: can you be more quantitative? For SrTiO₃, it is clear that there would be relatively large changes, but for sapphire, this is a very small change in permittivity. What about the expected change in resonator frequency due to metal expansion?

Response: The resonant frequency of SrTiO₃ changed mainly due to its high temperature coefficient of the permittivity, which leads to a large temperature coefficient of frequency of approximately 1700 ppm/K^{1,2}. Based on the volumetric thermal expansion coefficient of silver around $54 \times 10^{-6} \text{ K}^{-1}$, the temperature coefficient of resonant frequency of the all-metallic SRR cluster is estimated to be approximately 130 ppm/K. Therefore, the resonant frequency variation in the all-metallic structure due to thermal expansion is negligible, thanks to the good thermal conductivity and very low thermal coefficient of resonant frequency.

Revision: We have added the following text in the Results section:

- This excellent thermal stability is attributed to silver's high thermal conductivity and moderate thermal expansion coefficient, resulting in a temperature coefficient of resonant frequency of approximately 130 ppm/K. In contrast, the central frequency of the SrTiO₃ based maser shifts from 1.449 GHz to 1.453 GHz, resulting from SrTiO₃'s large temperature coefficient of resonant frequency (approximately 1700 ppm/K). We simulated the diffusion of the deposited optical pump energy and the resulting temperature variations in the SrTiO₃ resonator using COMSOL. The simulation results are presented in the Supplementary Movie1 and Fig.S6.

We have added Movie1 and Fig.S6 in the Supplementary materials.

R.3.7: 3. Mode volume should refer to the magnetic field in all directions (namely magnitude) in the denominator since it relates to total magnetic energy. What is $|H_{z,\text{max}}(r)|$ and how is it dependent on r ?

Response: Correct. However, since most of the magnetic field is concentrated in the core region of the resonator and mostly aligned along z axis, the mode volume was approximated using the z -component field H_z of the magnetic. $|H_{z_{\text{max}}}(r)|$ represents the maximum magnitude of the magnetic field in the resonator.

Revision: “Assuming that within this region, we align uniformly an ensemble of quantum magnetic emitters along the z -axis, we would get an effective magnetic mode volume $V_m = \int \frac{|H(r)|^2}{|H_{\text{max}}(r)|^2} dV$ of only 0.025 cm^3 .”

R3.8: 4. Figure 1 – can you plot the E-field as well?

Response: The E-field is shown below in Figure R3.8.1 and is also included in the revised supplementary material as Supplementary Fig. S2.

Figure R3.8.1 Electric field distribution within the cross section of the toroidal SRR cluster in the yz -plane (a) and xy -plane (b).

Revision: We have added this figure to the revised supplementary material as Supplementary Fig. S2.

R3.9: 5. Page 4 – it should be mentioned clearly that this maser operates only in pulsed mode. What about masers that need a static field? The all-metallic design with its shield is larger than the dielectric design, implying a larger and bulkier magnet.

Response: The all-metallic design might shield more static magnetic field than the dielectric design; however, this influence can be considered negligible due to the use of silver. To verify this, we conducted simulations in COMSOL to evaluate the impact of the all-metallic resonator. As shown in Figure R3.9.1, the presence of the SRR cluster made in silver has minimal impact on the static magnetic field distribution.

Figure R3.9.1 Static magnetic field between two electromagnets, simulated in COMSOL. a and b, Homogeneous distributions of the magnetic field in the center of the simulated region without and with the SRR cluster in silver, respectively. The black solid lines depict the magnetic field contours. **c,** Magnetic field profiles along the x axis at z=0mm from panel a and b, represented by a solid blue line and a dashed red line, respectively.

Revisions: In light of these promising simulation results, we have selected the pentacene-based **pulsed** maser as an ideal test platform, for which we can directly compare the performance obtained with our SRR cluster and with a high-index dielectric resonator.

R3.10: 6. Page 8 – what is the spin concentration of the sample?

Response: The spin concentration of the sample is approximately 1000 ppm, denoted as 0.1% mol/mol in the Method section on Page 8 of the manuscript.

References

1. Wise, P. L. *et al.* Structure-microwave property relations in $(\text{Sr}_x\text{Ca}(1-x))_{n+1}\text{Ti}_n\text{O}_{3n+1}$. *J. Eur. Ceram. Soc.* (2001).
2. Geyer, R. G., Riddle, B., Krupka, J. & Boatner, L. A. Microwave dielectric properties of single-crystal quantum paraelectrics KTaO_3 and SrTiO_3 at cryogenic temperatures. *J. Appl. Phys.* **97**, 104111 (2005).

3. Wu, H. *et al.* Enhanced quantum sensing with room-temperature solid-state masers. *Sci. Adv.* **8**, eade1613 (2022).
4. Breeze, J. *et al.* Enhanced magnetic Purcell effect in room-temperature masers. *Nat. Commun.* **6**, 6215 (2015).
5. Salvadori, E. *et al.* Nanosecond time-resolved characterization of a pentacene-based room-temperature MASER. *Sci. Rep.* **7**, 41836 (2017).
6. Anderson, C. P. & Awschalom, D. D. Embracing imperfection for quantum technologies. *Physics Today* **76**, 26–33 (2023).
7. Breeze, J. D., Salvadori, E., Sathian, J., Alford, N. McN. & Kay, C. W. M. Continuous-wave room-temperature diamond maser. *Nature* **555**, 493–496 (2018).

Response letter – 2nd review, manuscript number NCOMMS-23-52768

We would like to thank all the reviewers for their time and careful review of our manuscript. In the following, we provide a point-by-point response to each of the Reviewer's comments.

Reviewer 1:

R 1.1 The authors state that they do not observe any frequency shift using the all-metallic resonator. I find this comment confusing since they state later in the rebuttal they estimate the temperature coefficient of frequency to be 130 ppm/K. The shift here being due to thermal expansion of the silver. This is substantial and should be measurable at the frequency of the maser emission at 1.45 GHz.

Response: The theoretical temperature coefficient of 130 ppm/K reflects a worst-case scenario, assuming a uniform 1 K temperature rise throughout the silver structure. In practice, the actual temperature increase during operation is negligible due to the high thermal conductivity and efficient heat dissipation of the metallic resonator. Experimentally, we observed stable maser emission over 18,000 seconds without any detectable frequency drift, confirming that the real temperature variation was far below the threshold required to produce a measurable shift.

R 1.2 The authors have produced graphs showing temperature changes in the resonator. No detail on how these graphs were generated is given so I cannot comment further except to say that they authors did not properly address the original concerns.

Response: We appreciate the reviewer's feedback and acknowledge the need for clarity on how thermal effects were simulated. To directly address concerns about the thermal stability comparison between the SrTiO₃ resonator (with polystyrene foam support) and our metallic SRR design, we conducted COMSOL Multiphysics simulations to model heat accumulation and dissipation in the SrTiO₃ resonator with two support materials: sapphire and polystyrene foam.

Simulation setup:

1. Model Geometry (Figure 1.2.1):

- SrTiO₃ resonator: hollow cylindrical structure matching the experimental dimensions.
- Copper baseplate: circular disk with a central hole.
- Support: hollow cylinder made of either sapphire or polystyrene foam.
- Optical fiber end: modeled as a 1 mm-diameter disk positioned at the bore to represent the laser input.
- Default COMSOL material parameters were used for all components except the polystyrene foam, whose properties were obtained from:
<https://www.laird.com/sites/default/files/2021-01/RFP-DS-PP%2006242020.pdf> .

Figure R1.2.1 3D model in the COMSOL simulation

2. Physics

- Laser heating: simulated using the Geometrical Optics interface to trace 592 nm laser pulses (5 ns duration, 3 mJ energy, Gaussian profile).
- Heat transfer: modeled via the Heat Transfer in Solids interface to capture transient heat accumulation and dissipation.

3. Simulation protocol:

- Total duration: 5 s, divided into five 1 s cycles.
- Within each cycle:
 - First 5 ns: laser excitation and localized heating of the resonator.
 - Remaining time: passive cooling without laser input.
- Initial conditions:
 - Cycle 1 starts at 293.15 K (room temperature).
 - Each subsequent cycle begins with the final temperature distribution from the previous cycle.

R 1.3 Whilst an improved resonator temperature stability is admirable, I remain unconvinced that the work presented here is significant enough a development to warrant publication in Nature Communications. As pointed out by Reviewer 2, a patent was published in 2013 (WO 2013/175235 A1, see Figure 4) that describes using a split-ring resonator, very similar to that reported by the authors, suggesting that the idea is not original.

Response: We acknowledge that planar split-ring resonators (SRRs) were mentioned in the earlier patent WO 2013/175235 A1 as potential cavity architectures for maser applications. However, it does not demonstrate maser operation, Purcell enhancement, or any experimental validation. In general, we wish to point out that it is infinitely less-trivial to devise actual SRRs-based designs than just thinking that this might be possible, which is enough in a patent. We also wish to remark that patents do not automatically represent prior art, and may protect technical solutions that were never actually demonstrated and are extremely challenging to implement, providing they actually work.

In contrast, our study reports the first experimental realization of a room-temperature maser based on an all-metallic structure, supported by:

- comprehensive full-wave electromagnetic design, theory and demonstration
- quantitative analysis of Purcell enhancement, and
- evaluation of thermal robustness.

Moreover, the resonator described in the patent corresponds to a planar structure (~100 μm thick), while our device is a three-dimensional toroidal SRR cluster that achieves volumetric magnetic confinement. This architecture enables a deep-subwavelength mode volume and large Purcell factor, while maintaining exceptional thermal stability over thousands of optical excitation cycles. Based on our work, we do not believe that planar SRRs can reach sufficient Purcell factors over an appreciable gain medium volume: a volumetric design is required.

Thus, although conceptually related, the present work represents a substantial technical and experimental advance, establishing a clear and verifiable step forward in the development of metallic resonators for enhancing the emission rates of magnetic quantum emitters. We believe that rejection of our paper based on this patent would be scientifically unfounded.

Reviewer 2:

R2.0: The authors have addressed all my comments, and I now recommend the paper for publication.

Response: We sincerely thank the reviewer for their thorough evaluation and positive recommendation for publication.

Reviewer 3:

R3.0: The authors have made efforts to address the reviewers' comments and revise their manuscript accordingly. I see improvements; however, there are still several issues that I believe require further attention:

Response: We thank the reviewer for their constructive feedback and for recognizing the improvements in our revised manuscript. We appreciate the remaining points raised and have carefully addressed each of them in this version.

R3.1: (R3.3) As noted in my previous report, this work focuses on pulsed maser operation at a very low repetition rate (1 Hz), which has limited applicability. The authors provided in their response some estimates regarding the feasibility of supporting continuous-wave (CW) diamond NV-based masers—especially in relation to the resonator—but these were not incorporated into the revised paper.

Response: We thank the reviewer for highlighting this important point. Our focus on pulsed operation at 1 Hz in this work stems from the need to resolve the critical challenge of thermal instability in deep-subwavelength room-temperature masers for enabling more advanced applications. The present work therefore serves as a proof-of-principle demonstration of the resonator's thermal stability and Purcell enhancement under pulsed excitation. We fully agree that continuous-wave operation represents the next step toward broader applicability. Accordingly, we have now incorporated a discussion of CW feasibility into the revised manuscript and Supplementary Information, as detailed below.

Revision: We have inserted the following passage into the main manuscript:

In addition, the geometry of the clustered SRR can be tailored to achieve operation at different frequencies, accommodating other magnetic quantum emitters such as nitrogen–vacancy (NV) defect centers in diamond that are used in continuous-wave (CW) masers. A detailed analysis of the thermal stability of CW masers incorporating the SRR cluster is provided in Supplementary Discussion SII B.

Additionally, we included the following text in the Supplementary Material:

Supplementary Discussion SII

B - Thermal Stability for Continuous-Wave Maser Operation:

While the present work focuses on pulsed operation, the intrinsic thermal properties of the SRR cluster make it particularly promising for stable CW applications.

To evaluate the potential for continuous-wave (CW) operation, we analyzed the thermal performance of the silver toroidal SRR cluster and compared it with that of conventional dielectric resonators. Consider, for instance, a maser employing NV centers in diamond with a sapphire resonator, where continuous laser pumping at 400mW caused a frequency shift of 25MHz due to thermal heating. This corresponds to an estimated temperature rise: $\Delta T_{Al_2O_3} = 35^\circ C$.

Assuming that the toroidal SRR cluster has a similar size and is operated under the same conditions as the sapphire resonator, the temperature increase ΔT_{SRR} could be estimated as $\Delta T_{SRR}/\Delta T_{Al_2O_3} \sim \kappa_{Al_2O_3}/\kappa_{Ag} \approx 0.1$, following the Fourier's law, where $\kappa_{Al_2O_3}$ and κ_{Ag} are the thermal conductivities of sapphire and silver, respectively. Given that the volumetric thermal expansion coefficient of silver is approximately $54 \times 10^{-6} K^{-1}$, the geometric expansion would be less than 0.02%. Consequently, the frequency variation in the SRR cluster is expected to be negligible compared to that observed in the sapphire resonator.

R3.2: (R2.1) The discussion of the spontaneous emission rate and the Purcell factor is non-quantitative. Additionally, the relationship to the Einstein B-coefficient does not seem to be incorporated into the revised paper.

Response: At microwave frequency, spontaneous emission is negligible compared to stimulated emission, since $\frac{A_{21}}{B_{21}} = \frac{8\pi h\nu^3}{c^3}$, where A_{21} and B_{21} are the Einstein coefficients of spontaneous emission and stimulated emission, respectively, and ν is the photon frequency^{1,2}. Owing to the ν^3 dependence, the spontaneous emission rate is many orders of magnitude smaller than the stimulated emission rate in the GHz regime and can therefore be safely neglected in the maser dynamic model.

For the Purcell-enhanced stimulated emission, we have the following equations:

$$\begin{cases} \dot{q} = -k_c q + BNq \\ B = \frac{\mu_0 \hbar \omega \gamma_e^2 T_2}{4V_m} \\ k_c = \frac{\omega}{Q_{load}} \\ F_p = \frac{3\lambda_m^3}{4\pi^2} \cdot \frac{Q_{load}}{V_m} \end{cases}$$

Here, the Einstein coefficient of stimulated emission B scales inversely to the mode volume V_m , and the cavity photon decay rate k_c is inversely proportional to the loaded Q factor Q_{load} . The threshold of population inversion N_{th} for maser action is therefore

$$N_{th} = \frac{k_c}{B} \propto \frac{V_m}{Q_{load}} \propto \frac{1}{F_m}.$$

Hence, achieving a small mode volume and a high loaded Q-factor, corresponding to a large Purcell factor, is essential for low-threshold and efficient maser operation.

Revision: We have incorporated the quantitative relation between the Einstein B-coefficient, mode volume, and Purcell factor into the revised manuscript (Methods – Maser Dynamic Modelling) as follows:

B stands for the Einstein coefficient of the stimulated emission, which is directly linked to the resonator's mode volume as follows:

$$B = \frac{\mu_0 \hbar \omega \gamma_e^2 T_2}{4V_m}$$

where μ_0 is the vacuum permeability, \hbar the reduced Planck constant, $\gamma_e = 1.76 \times 10^{11} \text{ rad} \cdot \text{s}^{-1} \cdot \text{T}^{-1}$ the electron gyromagnetic ratio, and $T_2 = 2.1 \sim \mu\text{s}$ the spin-spin relaxation time of pentacene. Spontaneous emission is omitted from our dynamical model because, at microwave frequencies, its rate is vanishingly small compared to stimulated emission.

And:

Assuming steady-state operation ($\dot{q} = 0$), the threshold population inversion is given by: $N_{\text{threshold}} = k_c/B = 4.4 \times 10^{13}$. Since the Purcell factor scales as $F_p \propto \frac{Q_{\text{load}}}{V_m}$, it follows that $N_{\text{threshold}} \propto \frac{1}{F_p}$, confirming that an ultrasmall V_m and high Q_{load} (i.e. a large F_p) are key to achieving low-threshold, efficient maser operation.

R3.3: (R2.4) The definition of the mode volume used in the text is somewhat controversial in this context. Specifically, normalizing by the maximum of the magnetic field can produce artificially low “magnetic mode” volumes if there is a localized magnetic field “hot spot.” In contrast, if the microwave magnetic field is relatively homogeneous across the resonator volume, the chosen definition might be reasonable. However, a more relevant approach could be to define the 50% or 90% mode volume, referring to the region where 50% or 90% of the microwave magnetic energy is localized. This might explain the discrepancy between the measured data and the mode volume calculation (a factor of ~3). I believe the authors should discuss this point in more detail.

Response: In our study, the mode volume V_m was calculated within the resonator bore region occupied by the gain medium, as the Purcell enhancement depends on the spatial overlap between the resonator's magnetic field and the emitters' magnetic dipoles. Magnetic energy outside this region does not contribute to emission enhancement and is therefore excluded from the effective V_m relevant to maser operation. To verify that the simulated mode volume is not biased by field localization or numerical artifacts, we performed a statistical analysis of the magnetic energy density distribution inside the bore. The field exhibits moderate uniformity, with a coefficient of variation of 0.45 and a median-to-mean ratio of 0.94. The ratio of maximum to median magnetic energy density is 2.40, indicating that the field maximum is only modestly higher than the typical value and is not dominated by an isolated hotspot.

Furthermore, the minimal sub-volumes enclosing 50% and 90% of the magnetic energy are $V_{50} = 0.010\text{cm}^3$ and $V_{90} = 0.025\text{cm}^3$, corresponding to 31% and 78% of the bore volume ($V_{\text{bore}} = 0.032\text{cm}^3$), respectively. These results demonstrate that the magnetic energy is broadly distributed across the gain region, confirming that the simulated mode volume is not artificially reduced by localized peaks.

The simulations provide an idealized lower bound assuming perfect geometry and complete dipole alignment, whereas the experimental value naturally reflects structural imperfections, partial filling, and non-uniform emitter orientation. The resulting threefold difference between the simulated and experimental mode volumes is therefore reasonable and does not alter the physical interpretation of the Purcell enhancement. Together with the quantitative field-distribution metrics presented above, this agreement confirms that our estimation of V_m is physically meaningful and representative of the actual electromagnetic environment experienced by the emitters.

Revision: We have inserted the following passage into the discussion section of the main manuscript:

To confirm that the simulated mode volume is not artificially reduced by local field maxima, we quantitatively analyzed the magnetic energy distribution in the resonator bore region. The field was found to be broadly uniform without strong localization, as detailed in Supplementary Discussion SII A.

Additionally, we included the following text in the Supplementary Material:

Supplementary Discussion SII

A - Statistical Evaluation of the Mode Uniformity in the Resonator Bore Region

To evaluate whether the calculated mode volume is affected by localized field peaks or numerical artifacts, we performed a statistical analysis of the simulated magnetic-energy density within the bore region of the clustered SRR. The mode volume V_m was evaluated only within this region, since the Purcell enhancement depends on the overlap between the resonator's magnetic field and the emitters' dipoles, while magnetic energy outside the gain medium does not contribute to the emission process.

The simulated magnetic energy distribution exhibits moderate uniformity, with a coefficient of variation (CV) of 0.45 and a median-to-mean ratio of 0.94. The ratio of maximum to median magnetic-energy density is 2.40, showing that the field maximum is modestly higher than the typical value and not dominated by a single hotspot. Furthermore, the minimal sub-volumes enclosing 50% and 90% of the magnetic energy are $V_{50} = 0.010\text{ cm}^3$ and $V_{90} = 0.025\text{ cm}^3$, corresponding to 31% and 78% of the bore volume ($V_{\text{bore}} = 0.032\text{ cm}^3$), respectively. These results demonstrate that the magnetic energy is broadly distributed across the emitter region.

The simulation represents an idealized lower bound assuming perfect geometry and complete dipole alignment, whereas the experimental value includes realistic effects such as structural deformation, partial filling, and non-uniform emitter orientation. The resulting threefold difference

between the simulated (0.025 cm^3) and experimental (0.072 cm^3) mode volumes is therefore reasonable and consistent with realistic experimental conditions. Combined with the statistical metrics presented above, these results confirm that the mode volume estimation used in this work is physically meaningful and representative of the actual electromagnetic environment experienced by the emitters.

R3.4: (R2.5a) The size of the crystal and the specific dimensions of the device are crucial details that should be linked more explicitly to the required optimal resonator geometry. This appears to be another instance where the authors' response did not result in changes to the manuscript.

Response: To facilitate precise reproduction of our design, the 3D model of the resonator is now available as an STL file on Zenodo (DOI: <https://zenodo.org/doi/10.5281/zenodo.10012323>), which can be directly used for 3D printing. In addition, we have included the dimensions of the cylindrical gain medium in the revised manuscript.

Revision: We have expanded the introduction of the main manuscript as follows to link the size of practically grown gain media to that of a promising resonator:

However, as spin transition frequencies are commonly in the gigahertz range, i.e., with emission wavelengths λ_m on the order of tens of centimeters, leveraging the magnetic Purcell effect comes with the inherent challenge of dealing with the large ratio between λ_m^3 and achievable values of mode volume V_m . In practice, the physical dimensions of gain media that can be grown experimentally, such as organic crystals or diamond-based samples, occupy physical volumes that are roughly three orders of magnitude smaller than λ_m^3 (i.e., $\propto \text{cm}^3$). To address this mismatch, the resonator geometry must be engineered such that the region of strong magnetic field is confined to a volume only slightly larger than that of the gain medium, with its orientation optimized for alignment with the emitters' magnetic dipole moments. This "squeezing" of the magnetic mode maximizes spatial overlap with the active medium, thereby enhancing the efficiency of energy exchange.

Additionally, we have included the dimensions of the cylindrical gain medium:

As depicted in Fig.1.b, the pentacene:p-terphenyl (pc:pt) crystal (1.5 mm radius \times 10 mm height) is positioned within the bore of the clustered SRR, which is enclosed inside a copper cavity. The bottom 5 mm of the pc:pt crystal includes a central axial hole of 0.5 mm radius, whereas the top 5 mm remains solid.

R3.5: (R2.8) Here as well, I believe the text of the paper should be modified accordingly, reflecting the issues raised in the response.

Revision: We have replaced "thermal fluctuations" with "thermal variations" in the manuscript and revised the corresponding part of the introduction to clarify this point and avoid ambiguity as follows:

However, dielectric resonators with extremely high permittivities, such as SrTiO₃, are inherently sensitive to temperature variations due to their large temperature coefficient of permittivity. Even minor room temperature variations, or the unavoidable thermal heating coming from optical excitation or read-out, would inevitably shift the mode frequency away from the quantum emitter linewidth. As a result, the system must be continuously retuned, hindering stable operation and limiting the practicality of such dielectric resonators.

R3.6: It seems somewhat unusual to discuss frequency stability using MHz-scale data, presumably due to the short-pulse operation. Comparing this to the ~10 kHz linewidths of CW maser systems cited in the references is difficult. A more direct comparison or acknowledgment of the differences in operational modes would be helpful.

Response: We agree that the frequency-stability discussion in our pulsed maser system should not be directly compared to the narrow (~10 kHz) linewidths of continuous-wave (CW) masers. To clarify this distinction, we have added an explicit paragraph in Supplementary Discussion SII B explaining that our measurements characterize pulse-to-pulse spectral stability rather than absolute linewidth, and emphasizing the fundamental difference between the transient pulsed regime and the steady-state CW regime.

Revision: we included the following text in Supplementary Discussion SII B of the Supplementary Material:

While the present work focuses on pulsed operation, the excellent thermal properties of the SRR cluster make it particularly promising for stable continuous-wave (CW) applications. The observed MHz-scale linewidths in our pulsed maser experiments arise from the Fourier limit associated with microsecond pulse durations, rather than from intrinsic decoherence or thermal instability. In contrast, CW masers operate in a steady-state regime that can achieve linewidths on the order of tens of kilohertz. Although the absolute linewidths are therefore not directly comparable, the measurements presented here demonstrate exceptional pulse-to-pulse spectral stability of the SRR-based maser over prolonged operation times.

R3.7: (R3.1) The “quasi-2D” structures mentioned by the authors in relation to the quoted references of previous work have heights of approximately 0.5 mm and 1.4 mm—borderline for the intended application. However, there is no fundamental reason why these could not be extended to heights of, for example, 3 mm or more to fit the crystal used in this work. It is therefore unclear why the authors dismiss these structures as unsuitable for the current application.

Response: The reviewer is correct that quasi-2D resonator geometries could, in principle, be extended vertically. Our toroidal SRR cluster was inspired by planar SRR concepts but represents a conceptual three-dimensional evolution developed through detailed electromagnetic design to achieve genuine volumetric magnetic confinement. The resulting structure supports a uniform magnetic field across the crystal volume, exhibits deep-subwavelength compactness without incorporating any dielectric materials (unlike designs such as that in NCBI

<https://www.ncbi.nlm.nih.gov/pmc/articles/PMC6948142/> which rely on rutile loading), and maintains excellent thermal stability. This three-dimensional architecture differs fundamentally from a simple geometric extrusion and enables strong Purcell enhancement together with stable operation under repeated optical excitation.

R3.8: 4. (R3.3) The condition $C > 1$ is typically considered indicative of strong (rather than weak) coupling. Overall, while the revised manuscript addresses some of the reviewers' concerns, I believe further clarifications and revisions are needed in the areas detailed above.

Response: We thank the reviewer for pointing out the misuse and potential confusion in terminology. Our previous use of “relatively weak coupling” was intended to indicate a cooperativity value lower than those reported in other maser systems^{3,4}, rather than to suggest $C < 1$ in the strict formal sense. To avoid confusion, we have revised the manuscript accordingly, replacing “relatively weak coupling” with “strong coupling” to align with the conventional definition for $C > 1$.

Revision: We have replaced “relatively weak coupling” with “strong coupling” in the discussion section of the manuscript.

References

1. Einstein, A. Strahlungs-Emission und -Absorption nach der Quantentheorie. *Deutsche Physikalische Gesellschaft* **18**, 318–323 (1916).
2. Zangwill, A. *Modern Electrodynamics*. (Cambridge University Press, 2013).
3. Breeze, J. D., Salvadori, E., Sathian, J., Alford, N. McN. & Kay, C. W. M. Room-temperature cavity quantum electrodynamics with strongly coupled Dicke states. *Npj Quantum Inf.* **3**, 40 (2017).
4. Breeze, J. D., Salvadori, E., Sathian, J., Alford, N. McN. & Kay, C. W. M. Continuous-wave room-temperature diamond maser. *Nature* **555**, 493–496 (2018).